# Proof of crystal-field-perturbation-enhanced luminescence of lanthanide-doped nano-crystals through interstitial H$^+$ doping

Guowei Li [1,2], Shihui Jiang[1,2], Aijun Liu[1], Lixiang Ye[1,2,3], Jianxi Ke[1,4], Caiping Liu[1,4,5], Lian Chen [1] ✉, Yongsheng Liu [1,2,4,5] ✉ & Maochun Hong [1,2,4,5] ✉

Crystal-field perturbation is theoretically the most direct and effective method of achieving highly efficient photoluminescence from trivalent lanthanide (Ln$^{3+}$) ions through breaking the parity-forbidden nature of their 4$f$-transitions. However, exerting such crystal-field perturbation remains an arduous task even in well-developed Ln$^{3+}$-doped luminescent nanocrystals (NCs). Herein, we report crystal-field perturbation through interstitial H$^+$-doping in orthorhombic-phase NaMgF$_3$:Ln$^{3+}$ NCs and achieve a three-orders-of-magnitude emission amplification without a distinct lattice distortion. Mechanistic studies reveal that the interstitial H$^+$ ions perturb the local charge density distribution, leading to anisotropic polarization of the F$^-$ ligand, which affects the highly symmetric Ln$^{3+}$-substituted [MgF$_6$]$^{4-}$ octahedral clusters. This effectively alleviates the parity-forbidden selective rule to enhance the 4$f$–4$f$ radiative transition rate of the Ln$^{3+}$ emitter and is directly corroborated by the apparent shortening of the radiative recombination lifetime. The interstitially H$^+$-doped NaMgF$_3$:Yb/Er NCs are successfully used as bioimaging agents for real-time vascular imaging. These findings provide concrete evidence for crystal-field perturbation effects and promote the design of Ln$^{3+}$-doped luminescent NCs with high brightness.

The use of trivalent lanthanide (Ln$^{3+}$)-doped inorganic luminescent nanocrystals (NCs) as bioimaging agents capable of deep-tissue penetration and therapeutic agents for diverse critical diseases has come to the forefront through interdisciplinary research over the past decade[1–3]. By synchronously generating upconversion luminescence (UCL) and near-infrared (NIR) luminescence upon single-wavelength NIR irradiation (for example, using a 980 nm diode laser)[4–6], Ln$^{3+}$-doped luminescent NCs facilitate in vivo deep-tissue bioimaging with several inherent advantages over other fluorescent bioimaging agents[1,7–9], including clinically used indocyanine green (ICG). These

advantages include high temporal and spatial resolution and being non-photobleaching, even over long periods[2,10–12]. Nevertheless, the clinical application of Ln$^{3+}$-doped inorganic NCs is limited by their low emission brightness[6,13], particularly under complex physiological conditions. To address this limitation, several photoluminescence (PL) enhancement strategies such as core−shell passivation have been developed[12,14–17]. However, these approaches introduce engineering complexity. Moreover, overcoming this intrinsic limitation fundamentally is important for the further development of this significant class of materials. The fundamental cause of the limited PL brightness

[1]State Key Laboratory of Structural Chemistry, Fujian Institute of Research on the Structure of Matter, Chinese Academy of Sciences, Fuzhou, China. [2]University of the Chinese Academy of Sciences, Beijing, China. [3]Fujian Center for Safety Evaluation of New Drug, Fujian Medical University, Fuzhou, China. [4]Fujian Science & Technology Innovation Laboratory for Optoelectronic Information of China, Fuzhou, China. [5]Advanced Energy Science and Technology Guangdong Laboratory, Huizhou, China. ✉e-mail: cl@fjirsm.ac.cn; liuysh@fjirsm.ac.cn; hmc@fjirsm.ac.cn

of Ln$^{3+}$-doped NCs, as described by the Judd–Ofelt theory[18,19], is the low electric dipole transition probability due to the parity-forbidden nature of 4$f$-transitions of Ln$^{3+}$ emitters. Notably, a small deviation from the equilibrium symmetry of the Ln$^{3+}$ emitter during vibrational motion can lead to additional ligand–field interactions[20–23]. This promotes the mixing of even-parity 4$f$ configurations with their opposite-parity counterparts and thereby improves the efficiency of Ln$^{3+}$-emitter intra-4$f$ optical transitions[24,25]. This phenomenon, known as crystal-field perturbation, holds great promise for achieving ultra-bright Ln$^{3+}$-doped inorganic NCs without the need for crystal enlargement.

Impurity doping has been extensively investigated as a simple means of exerting crystal-field perturbation on the Ln$^{3+}$ emitters in Ln$^{3+}$-doped luminescent NCs[26–28]. Dopant ions with large size- and valence state-mismatch to the host atoms cause local distortion, thus reducing the site symmetry of the Ln$^{3+}$ emitter[27–29]. This local symmetry-breaking should facilitate intra-4$f$ optical transitions and increase the radiative transition rate, leading to enhanced PL emission and a shortened PL lifetime. To achieve this, Li$^+$ ions, which are much smaller than the host cations, have been introduced into the host lattices of the well-developed Ln$^{3+}$-doped ALnF$_4$-type (A = Na, K) inorganic fluoride NCs[27,30]. However, in most previous studies, the PL lifetime of the Ln$^{3+}$ emitter was prolonged[21,28,31]. This is because doping with Li$^+$ increases the crystal size[27,28], changing the crystallinity[32] and even the physical phase of the host lattice[30], which may be responsible for the reduced nonradiative transition effect along with the extended PL lifetime. Consequently, no genuine crystal-field perturbation effect is exerted on the Ln$^{3+}$ activator. To directly prove that crystal-field perturbation can improve the PL efficiency of Ln$^{3+}$ emitters, the overall crystallographic structure must remain essentially unchanged to exclude other influencing factors, which hitherto has remained an arduous task.

H$^+$ ions have very small radii and high chemical activity; therefore, they are readily incorporated into the interstitial sites of inorganic crystals without affecting the crystal structure[33,34]. To date, H$^+$-ion doping has been widely studied in electrochemistry[35,36]. For example, Vanka et al.[36] reported that interstitial H$^+$-doping reduces the potential barrier and tunes the surface resistance of SrTiO$_3$. Additionally, Nakayama et al.[33] concluded that interstitial H$^+$-doping holds great promise for improving the conductivity of ZrO$_2$ semiconductors without affecting the crystal structure. Inspired by these results, we introduced H$^+$ ions into Ln$^{3+}$-doped alkali metal fluoride NCs with high symmetry (e.g., $S_6$ or $O_h$) to stimulate crystal-field perturbation in its truest sense. The dopant H$^+$ ions have negligible influence on the crystal structure parameters, yet they disturb the local charge density distribution and contribute to the anisotropic polarization of the ligand (F$^-$) of the Ln$^{3+}$ emitter[37–39]. This introduces an additional field in the local structure around the Ln$^{3+}$ emitter and facilitates the mixing of odd-parity states into the 4$f$ wavefunction[22,23,38]. According to the Judd–Ofelt principle[18,19,38,40,41], the mixing of opposite-parity configurations enhances the oscillator strength of the electric dipole transition ($f_{ed}$) of Ln$^{3+}$ emitter[38]. The electric dipole radiative transition probability ($A_{ed}$) is positively correlated with $f_{ed}$ ($A_{ed} \propto f_{ed}$), such that an elevation of $f_{ed}$ is accompanied by an increase in the $A_{ed}$ of intra-4$f$ optical transitions. This ultimately transforms the dim luminescence emission caused by high crystal symmetry to bright luminescence emission (Fig. 1a and Supplementary Methods).

## Results

### Experimental model selection and basic characterization of crystal-field perturbation

In our design, an orthorhombic NaMgF$_3$ crystal was selected as a model host matrix to prepare interstitially H$^+$-doped NaMgF$_3$:Yb/Er NCs. Based on preliminary studies, the nominal additions of Yb$^{3+}$ and Er$^{3+}$ were set to 4 and 1 mol%, respectively, to optimize the PL intensity while maintaining the pure orthogonal NaMgF$_3$ phase (Supplementary

Fig. 1). NaMgF$_3$ has rich lattice voids that are hypothesized to facilitate interstitial H$^+$-doping (Fig. 1b, Supplementary Table 1 and Supplementary Fig. 2). This reasoning was supported by first-principles density functional theory (DFT) calculations, which demonstrated that interstitial H$^+$-doping in orthorhombic NaMgF$_3$ is relatively easier than other doping behaviors, such as substitutional H$^+$-doping (Fig. 1c and Supplementary Fig. 3). Motivated by these positive DFT results, we synthesized a series of interstitially H$^+$-doped NaMgF$_3$:Yb/Er NCs using a modified high-temperature coprecipitation method in a nitrogen atmosphere, whereby H$^+$ ions were intentionally introduced by adding acetic acid (HAc) to the reaction environment (see Methods and Supplementary Figs. 4 and 5). For simplicity, we denote the as-synthesized NaMgF$_3$:Yb/Er NCs as NMF-H-X, where X is the nominal amount of HAc precursor (0, 1.6, 2.6, 3.1, 7.3, and 14.7 mmol).

The as-synthesized NMF-H-X NCs had nearly identical tetragonal shapes (Fig. 2a). NMF-H-0, NMF-H-1.6, NMF-H-2.6, NMF-H-3.1, NMF-H-7.3, and NMF-H-14.7 had average sizes of 12.5 ± 1.5, 6.7 ± 0.6, 9.0 ± 0.8, 12.3 ± 1.1, 16.9 ± 1.5, and 18.3 ± 1.5 nm, respectively (Fig. 2a and Supplementary Fig. 6). Powder X-ray diffraction (XRD) patterns of all the NCs were indexed to orthorhombic NaMgF$_3$ (space group $Pbnm$ (No. 62), $(a, b, c)$ = (5.365, 5.492, 7.674 Å), $Z$ = 4; JCPDS No. 13-0303), confirming the formation of highly crystalline NaMgF$_3$:Yb/Er (Fig. 2b, left). Notably, no appreciable peak shifting was detected upon H$^+$ doping (Fig. 2b, middle). This differs fundamentally from the case of other cation-doped inorganic NCs, such as Gd$^{3+}$-doped NaYF$_4$:Yb/Er NCs, which undergo significant structural distortion and even phase changes with increasing cation concentration[26]. Consequently, this finding strongly supports the notion that interstitial H$^+$-doping only exerts a minor effect on the lattice parameters of NaMgF$_3$:Yb/Er (i.e., crystal-field perturbation). This was further confirmed by the minor change in cell volume after interstitial H$^+$-doping, as derived from high-resolution XRD fitting and first-principles calculations (Fig. 2b and Supplementary Tables 2 and 3).

High-angle annular dark-field scanning transmission electron microscopy (HAADF-STEM) and two-dimensional energy-dispersive X-ray spectroscopy (EDS) elemental mapping of randomly selected NMF-H-X NCs revealed that Yb$^{3+}$ and Er$^{3+}$ were homogeneously distributed (Fig. 2c), verifying the successful doping of Yb$^{3+}$ and Er$^{3+}$ into the orthorhombic NaMgF$_3$ matrix. Compositional analysis using EDS and inductively coupled plasma optical emission spectrometry (ICP-OES) further corroborated the presence of host elements Na, Mg, and F and dopants Yb and Er in the as-synthesized NMF-H-X NCs (Supplementary Fig. 7 and Supplementary Table 4). Although H$^+$ was not detectable by EDS or ICP-OES owing to its low relative atomic mass, successful interstitial H$^+$-doping was well proven by first-principles calculations, because the formation of interstitially H$^+$-doped NaMgF$_3$:Yb/Er is energetically favorable to that of NMF-H-0 (Fig. 1c and Supplementary Fig. 3).

### Gain effect of crystal-field perturbation on upconversion optical response

To understand the effect of interstitial H$^+$-doping on the optical behavior of Er$^{3+}$ emitters in the NMF-H-X NCs, we measured the room-temperature UCL upon excitation with a 980 nm diode laser (power density: ~50 W/cm$^2$). The NMF-H-X NCs exhibited a dominant red UCL band at ~654 nm and a secondary green emission band at ~545 nm (Fig. 3a), which correspond to the $^4F_{9/2} \rightarrow {}^4I_{15/2}$ and $^2H_{11/2},{}^4S_{3/2} \rightarrow {}^4I_{15/2}$ transitions of Er$^{3+}$, respectively. The calculated red-to-green UCL intensity ratios of Er$^{3+}$ in the NMF-H-X NCs were as high as 15.6–45.8 (Supplementary Fig. 8), yielding a series of bright-red outputs (inset of Fig. 3a). Notably, despite the low Yb/Er content (≤2.1 mol%, Supplementary Table 4), the overall upconversion quantum yield (UCQY) of the interstitially H$^+$-doped NaMgF$_3$:Yb/Er NCs increased from <0.01% to 0.18% in the presence of interstitial H$^+$ (Supplementary Table 5), resulting in luminescence that was visible to the naked eye. Indeed, the

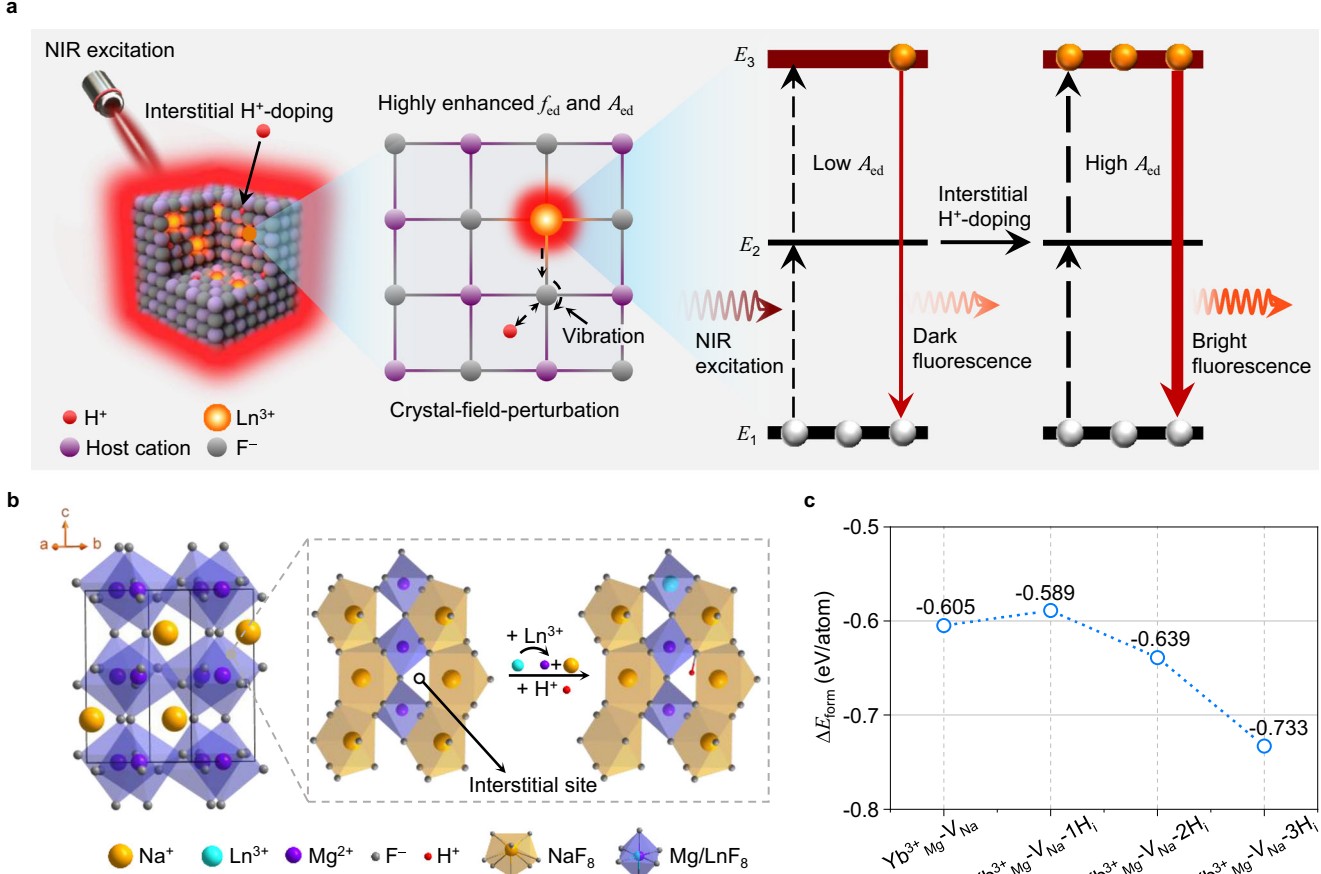

**Fig. 1 | Theory of crystal-field-perturbation and upconversion luminescence (UCL) enhancement through interstitial H⁺-doping of orthorhombic NaMgF₃:Yb/Er nanocrystals (NCs).** **a** Schematic illustrations of interstitial H⁺-doping (left), crystal-field perturbation (middle), and UCL enhancement (right). When H⁺ ions are doped into interstitial lattice sites, they form hydrogen bonds with the neighboring F⁻ ions, perturbing the crystal field around the Ln³⁺ ions and relaxing the Ln–F bonds, and thereby mitigating the parity-selection rule of the Ln³⁺ emitter. Thus, under near-infrared (NIR) irradiation, the $f_{ed}$ and $A_{ed}$ are greatly enhanced. This means that during the upconversion transition process, the probability of the electrons of Ln³⁺ to undergo the intra-4$f$ transition from the ground state ($E_1$) to the intermediate level ($E_2$) and then to the excited state ($E_3$), as well as the radiation transition probability of electrons falling back from $E_3$ to $E_1$, will be increased, thus greatly amplifying the luminescence. **b** Crystal structure and doping sites of orthorhombic NaMgF₃:Yb/Er NCs. In this system, Ln³⁺ ions substitute Mg²⁺ ions at the octahedral symmetry centers ($S_6$), and H⁺ ions are doped in interstitial lattice sites. **c** Formation energy ($\Delta E_{form}$) per atom for NaMgF₃:Yb³⁺$_{Mg}$-$V_{Na}$-$x$H$_i$ NCs as a function of the number of H⁺ interstitial defects ($x$), as calculated by first-principles density functional theory (DFT). Yb³⁺$_{Mg}$ represents Yb³⁺ in a Mg²⁺ site, $V_{Na}$ is a Na⁺ vacancy, and H$_i$ is interstitial H⁺. $\Delta E_{form}$ is the energetic difference between the NaMgF₃:Yb³⁺$_{Mg}$-$V_{Na}$-$x$H$_i$ NCs and the isolated constituent atoms. Source data are provided as a Source Data file.

overall UCL intensity of the NMF-H-14.7 NCs was up to 675 times higher than that of NMF-H-0 (Fig. 3b and Supplementary Fig. 9), with green and red UCL intensity enhancement factors of 258 and 706.3, respectively (Supplementary Fig. 10). Even for the smallest interstitially H⁺-doped NCs (NMF-H-1.6; mean crystal size: 6.7 ± 0.6 nm), the UCL intensity was approximately twice that of NMF-H-0 (12.5 ± 1.5 nm), despite the halving of crystal size (Fig. 3b). For the NMF-H-3.1 and NMF-H-0 NCs, which had a similar particle size (~12 nm), interstitial H⁺-doping increased the fluorescence intensity by a factor of 85. In addition, the fluorescence intensities of the NMF-H-7.3 and NMF-H-14.7 NCs were 245.6 and 159.8 times greater than those of NMF-H-0 samples with equivalent particle sizes (~16 and 18 nm, respectively; Supplementary Fig. 11). This demonstrates that the enhanced UCL intensity of the Er³⁺ emitter was not solely caused by crystal size enlargement. In fact, the UCL intensity was enhanced even without increasing the crystal size, which differs markedly from typical PL enhancement approaches, such as core–shell engineering.

To further exclude the effect of crystal size on the UCL intensity, we synthesized two sets of core–shell NCs in which NMF-H-0 and NMF-H-14.7 were utilized as core seeds with inert NaMgF₃ coatings of different thicknesses (Supplementary Fig. 12). The

core–shell NMF-H-0@NaMgF₃ NCs, similar to previously reported core–shell NCs, exhibited increased UCL intensity and lifetime owing to the inert shell coating (Fig. 3c and Supplementary Fig. 13). By contrast, the UCL intensity of the core–shell NMF-H-14.7@NaMgF₃ NCs gradually decreased with increasing shell thickness, accompanied by a stepwise increase in the UCL lifetime (Fig. 3d and Supplementary Fig. 14), which indicates a completely different mechanism of Er³⁺ UCL enhancement. This may stem from the diffusion of H⁺ ions from the NMF-H-14.7 core to the pure NaMgF₃ shell owing to the high chemical activity of H⁺ ions and lower formation energy for interstitial H⁺-doping in pure NaMgF₃ than in NaMgF₃:Yb/Er (Supplementary Fig. 3). Such H⁺-ion diffusion would weaken the crystal-field perturbation effect in the NMF-H-14.7 core. To investigate this hypothesis, we prepared core–shell NMF-H-7.3@NaMgF₃:H NCs in which a similar concentration of H⁺ was doped into the NaMgF₃ shell as that in the NMF-H-7.3 core, with the aim of suppressing the diffusion of H⁺-ions from the H⁺-doped core to the pure NaMgF₃ shell. The UCL intensity of the NMF-H-7.3 NCs was enhanced by a thin H⁺-doped NaMgF₃ shell (Supplementary Fig. 15), corroborating the hypothesis that the reduction in UCL intensity with increasing shell thickness for the NMF-H-

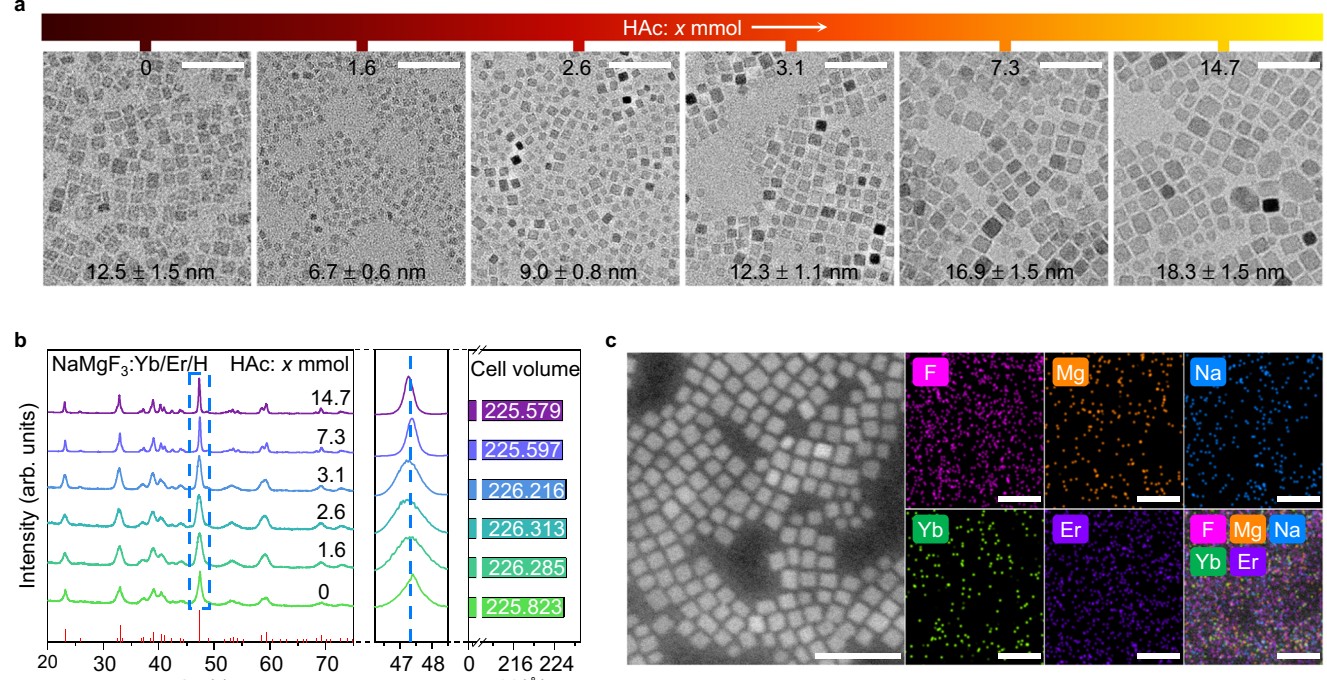

**Fig. 2 | Morphological, structural, and compositional characterization of NaMgF₃:Yb/Er (Yb/Er = 4/1 mol%) (NMF-H-X) nanocrystals (NCs) with and without interstitial H⁺-doping. a** Transmission electron microscopy (TEM) images and **b** high-resolution X-ray diffraction (XRD) patterns of the NMF-H-X NCs with and without interstitial H⁺-doping as a function of the nominal amount of acetic acid (HAc) used in the synthetic procedure (X = 0–14.7 mmol, the color bar from dark red to bright yellow in **a** represents an increase in the nominal addition of HAc). The NC sizes are listed at the bottom of each panel in **a**. The middle and right panels of **b** show the enlarged angular region of 46.2°–48.5° and the cell volumes, respectively. **c**, High-angle annular dark-field-scanning transmission electron microscopy (HAADF-STEM) image of randomly selected NMF-H-3.1 NCs and corresponding two-dimensional energy-dispersive X-ray spectroscopy (EDS) elemental mapping images of Na, Mg, F, Yb, and Er. Scale bars in **a** and **c**: 50 nm. Source data are provided as a Source Data file.

14.7@NaMgF₃ NCs was due to H⁺-ion diffusion. This further illustrates the important role of H⁺ in NaMgF₃:Yb/Er crystals.

Interstitial H⁺-doping was achieved by adding HAc as a H⁺ source during synthesis. To exclude the effect of acetate anions (Ac⁻) on the UCL intensity enhancement, we modified the synthesis of the NMF-H-0 NCs (see Methods) to use an Ac⁻-containing Na⁺ source (sodium acetate (NaAc)) instead of the original sodium hydroxide (NaOH). Thus, we prepared NaMgF₃:Yb/Er NCs without H⁺-doping but with Ac⁻ ions. As expected, the UCL intensities of the NMF-H-0 NCs synthesized with NaOH and NaAc were similar (Fig. 3e and Supplementary Fig. 16), confirming that Ac⁻ anions were not responsible for the significant UCL intensity enhancement shown in Fig. 3a and Supplementary Fig. 16. Notably, introducing H⁺ ions to the NMF-H-X NCs synthesized with NaAc resulted in a gradual increase in UCL intensity (Supplementary Fig. 17a), although the synthesized samples had a worse crystal morphology than those synthesized with NaOH (Supplementary Fig. 17b). The luminescence of the NMF-H-X NCs synthesized with NaOH was also not affected by acid washing to remove surface ligands (Supplementary Fig. 18). These experiments corroborate the theory that the UCL intensity enhancement was directly caused by the crystal-field perturbation exerted by interstitial H⁺-doping.

## Mechanism of crystal-field perturbation by interstitial H⁺-doping

Solid-state nuclear magnetic resonance (SSNMR) is a high-precision characterization technique that provides critical information on the local structure around specific atoms or ions. Notably, it can be used to demonstrate effective doping and bond formation[21,42–44]. The NMF-H-0 and NMF-H-3.1 NCs have the same particle size and, therefore, similar interfacial environments; thus, the successful interstitial H⁺-doping and formation of F–H···F bonds were well rationalized by the

noticeable chemical shift (by -1.55 ppm) in the ¹⁹F-SSNMR spectrum (Supplementary Fig. 19). The H⁺ ions occupy interstitial lattice sites near unsubstituted or Ln³⁺-substituted [MgF₆]⁴⁻ octahedra in the orthorhombic NaMgF₃ matrix (Fig. 4a). Owing to the strong electronegativity of F⁻ ions, the interstitial H⁺ ions likely form hydrogen bonds with the nearest neighboring F⁻ ions in the [MgF₆]⁴⁻ octahedra (i.e., to form F–H···F bonds), as clearly shown in the calculated electron localization function diagram, electronic band structures, and density of states (Supplementary Figs. 20 and 21). The formation of F–H···F bonds was further confirmed by the slight upshift (by -0.39 eV) in the binding energy of the X-ray photoelectron peak of the 1s state of F⁻ anions after interstitial H⁺-doping (Fig. 4b). The F–H···F bonds exert a slight effect on the local coordination environment around Ln³⁺ in the substituted [MgF₆]⁴⁻ octahedra—that is, crystal-field perturbation—by nonequivalently changing the bond lengths, angles, (Fig. 4c), and differential charge density distributions (Fig. 4d). This implies that the ligands (F⁻) of the Ln³⁺ ions are anisotropically polarized in the presence of interstitial H⁺ ions. The oscillating dipoles induced in the local structure lead to additional ligand–field interactions that promote the mixing of opposite-parity states of Ln³⁺, thus partially breaking the parity-forbidden nature of the intra-configurational 4f transitions[37,38]. The change in the transition dipole moment further proves this (Supplementary Fig. 21). Consequently, $A_{ed}$ increases, which means the activator is more likely to transition from the emitting state to a specific low-lying energy level, resulting in a significant enhancement in the UCL intensity (Fig. 1a).

The crystal-field-perturbation-enhanced $A_{ed}$ was directly supported by UCL decay experiments in which the red UCL band at ~654 nm was monitored during 980 nm pulsed laser excitation under cryogenic conditions (10 K). The UCL lifetime of the ⁴F₉/₂ state of the Er³⁺ emitter in NMF-H-3.1 was approximately one-third that in the NMF-

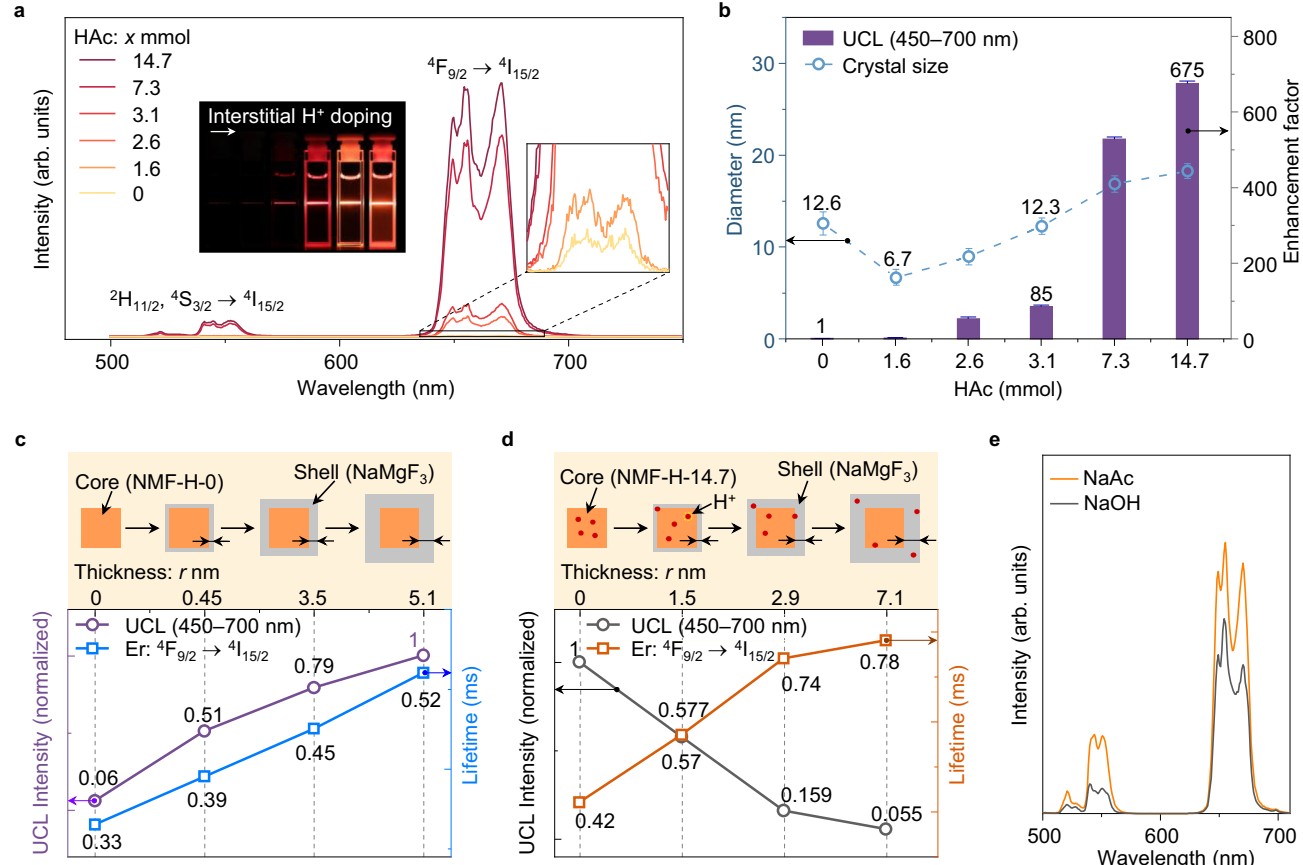

**Fig. 3 | Upconversion luminescence (UCL) intensity enhancement by interstitial H⁺-doping and exclusion of crystal size and acetate anion (Ac⁻) effects. a** Typical UCL spectra of as-synthesized NaMgF$_3$:Yb/Er NCs (NMF-H-X) nanocrystals (NCs) with different crystal sizes as a function of the nominal amount of acetic acid (HAc) added during the synthetic procedure (X = 0–14.7 mmol). The inset shows corresponding photoluminescence (PL) images collected using a Canon digital camera. **b** Evolution of UCL intensity ratio for NMF-H-X NCs and their corresponding crystal sizes as a function of the nominal amount of HAc. The error bars represent the standard deviation of luminescence enhancement factor (purple) and crystal size (blue). **c, d** Schematic illustration of NMF-H-0@NaMgF$_3$ (**c**) and NMF-H-14.7@NaMgF$_3$ (**d**) core–shell NCs with different thickness inert NaMgF$_3$ shells, and the corresponding Er³⁺ UCL intensity (purple/black line, normalized) and lifetime (blue/orange line) of the $^4F_{9/2}$ state when excited with a 980 nm laser. **e** UCL spectra of NaMgF$_3$:Yb/Er NCs (without H⁺-doping) synthesized with NaOH (black line) and NaAc (orange line) precursors. The spectra were collected under 980 nm excitation. Source data are provided as a Source Data file.

H-0 reference (0.21 vs. 0.6 ms, Fig. 4e). According to the Judd–Ofelt theory, the observed lifetime ($\tau$) of a particular excited state of an Ln³⁺ emitter can be calculated as $\tau = (A_{ed} + W_{NR})^{-1}$, where $W_{NR}$ is the non-radiative transition probability[5,45]. Considering that the NMF-H-0 and NMF-H-3.1 NCs have identical sizes and shapes, it is reasonable to assume that their $W_{NR}$ values are approximately equal. Therefore, the shortened UCL lifetime of Er³⁺ in NMF-H-3.1 (Fig. 4e and Supplementary Fig. 22) strongly supports the notion that $A_{ed}$ was enhanced. Together with the simultaneously increased ability of Yb³⁺ ions to absorb 980 nm excitation light (Fig. 4f), we detected a relatively stronger red UCL intensity in the interstitially H⁺-doped NaMgF$_3$:Yb/Er NCs (Figs. 1a and 3a).

To gain more insight into the crystal-field perturbation effect exerted by interstitial H⁺-doping, we studied the NCs using extended X-ray absorption fine structure (EXAFS) spectroscopy (Fig. 4g and Supplementary Figs. 23 and 24). The results showed that interstitial H⁺-doping had a negligible effect on the average Yb–F interatomic distance ($R_{Yb-F}$) and first shell coordination number of the Yb³⁺ ions (Fig. 4g and Supplementary Table 6), which is consistent with our high-resolution XRD observations (Fig. 2b). Additionally, first-principles calculations demonstrated that interstitial H⁺ ions only perturb the F–Ln–F bond angle of the [YbF$_6$]³⁻ sublattice structure, while the Yb–F bond length remains unchanged (Supplementary Table 7 and Supplementary Fig. 25). The similarity in the temperature-dependence of

the UCL spectra of the NMF-H-0 and NMF-H-3.1 NCs (Supplementary Fig. 26) indicates that Er³⁺ remains at the center of the highly symmetric [MgF$_6$]⁴⁻ octahedra, which further confirms the crystal-field perturbation effect exerted by interstitial H⁺-doping.

## Generalizability and applicability of interstitial H⁺-doping strategy

To elucidate the generalizability of the interstitial H⁺-doping strategy for enhancing the UCL of Ln³⁺ ions, we prepared NaMgF$_3$:Yb/Ho and NaMgF$_3$:Yb/Tm NCs (i.e., with different Ln³⁺ emitters) under synthetic conditions identical to those of the NaMgF$_3$:Yb/Er NCs (see Methods), with nominal additions of 14.7 mmol HAc to achieve interstitial H⁺-doping. The overall UCL intensities were significantly enhanced by interstitial H⁺-doping, with maximum UCL enhancement factors of approximately 482 and 362, respectively (Fig. 5a and Supplementary Fig. 27), corroborating the generalizability of our proposed interstitial H⁺-doping strategy. In another set of experiments, we prepared interstitially H⁺-doped NaMgF$_3$:Yb/Er NCs with different H⁺ sources (formic acid (HCOOH), hydrochloric acid (HCl), propionic acid (PA), and benzenesulfonic acid (BL70; PhSO$_3$H); see Methods). Er³⁺ UCL enhancements were achieved in all cases (Supplementary Figs. 28–31), with the BL70 precursor providing the largest enhancement factor of 1891 (Fig. 5b, Supplementary Fig. 31 and Supplementary Note 1). This was hypothesized to be because the benzenesulfonate anion (PhSO$_3^-$)

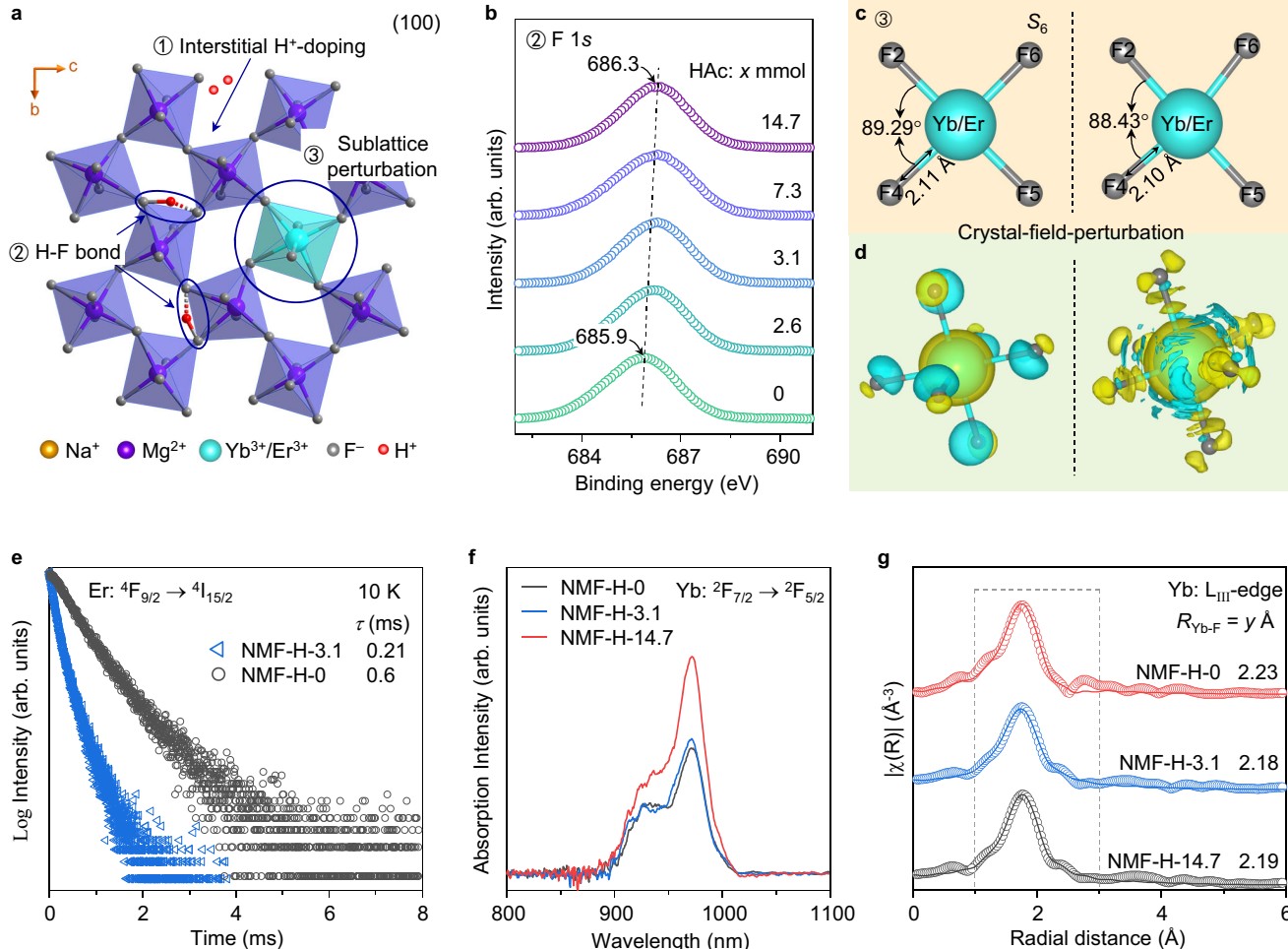

**Fig. 4 | Confirmation of hydrogen bond formation and resulting crystal-field-perturbation upon interstitial H+-doping. a** Schematic illustration of interstitial H+-doping of the NaMgF₃:Yb/Er lattice to form stable hydrogen bonds (F−H···F) that result in crystal-field perturbation. **b** High-resolution X-ray photoelectron spectroscopy (XPS) (F 1$s$) profiles of NaMgF₃:Yb/Er (NMF-H-X) nanocrystals (NCs) with different nominal acetic acid (HAc) additions (X = 0–14.7 mmol). The shift in peak position with increasing HAc addition demonstrates the formation of hydrogen bonds. **c, d** Crystal-field perturbations lead to limited changes in the bond lengths and bond angles within [ErF₆]³⁻ or [YbF₆]³⁻ (**c**) as well as significant changes in the differential charge density distribution (**d**). **e** Comparison of the $^4F_{9/2}$ lifetime ($\tau$) of the Er³⁺ emitters in the NMF-H-0 and NMF-H-3.1 NCs at 10 K. **f** Comparison of the typical absorption spectra of NMF-H-0, NMF-H-3.1, and NMF-H-14.7 NCs at 972 nm ($^2F_{7/2} \rightarrow {}^2F_{5/2}$ transition of Yb³⁺). **g** Experimental (dots) and Fourier-transform fitting results (solid lines) of Yb L$_{III}$-edge EXAFS spectra of NMF-H-0, NMF-H-3.1, and NMF-H-14.7 NCs, confirming that the structure of the NaMgF₃:Yb/Er NCs was essentially unchanged after H+-doping. $R_{Yb-F}$ is the average Yb−F interatomic distance. Source data are provided as a Source Data file.

has a larger molecular mass than the other precursor anions and therefore facilitates interstitial H+-doping. Interestingly, when the conjugated bases corresponding to these acids were used as Na⁺ sources, they deteriorated the crystal morphology, whereas using the acids as H⁺ sources stabilized the crystal morphology (Supplementary Fig. 32). Importantly, the interstitial H+-doping strategy was also suitable for other host matrices with high symmetry, such as CaF₂ and cubic α-NaYF₄, with overall UCL intensity enhancement factors of 807.9 and 399 for CaF₂:Yb/Er and α-NaYF₄:Yb/Er, respectively (Supplementary Figs. 33 and 34). The UCQYs increased from 0.015 and 0.037 to 0.514 and 0.16, respectively (Supplementary Table 8 and Supplementary Note 2). However, the high-temperature metastable cubic phase of α-NaYF₄:Yb/Er changes to the high-temperature stable hexagonal phase when using a large nominal amount of HAc (Supplementary Fig. 34), which is likely due to the large number of interstitial atomic defects[46]. Interestingly, after complete conversion to hexagonal β-NaYF₄:Yb/Er, the continued addition of HAc to 19.2 mmol enabled a 10-fold increase in UCL intensity to ultimately obtain bright β-NaYF₄:Yb/Er NCs with a UCQY of 2.6% (Supplementary Fig. 34 and Supplementary Table 8).

The interstitial H+-doping strategy is also effective for enhancing the downconversion luminescence (DCL) of Er³⁺ in the second near-infrared (NIR-II) spectral region (1000–1700 nm). As shown in Fig. 5c, d, increasing the nominal amount of HAc from 0 to 14.7 mmol increased the NIR-II luminescence intensity of Er³⁺ at 1532 nm by 259-fold, enabling the interstitially H+-doped NaMgF₃:Yb/Er NCs to serve as ideal contrast agents for high-resolution NIR-II angiography. Moreover, these interstitially H+-doped NCs have good crystal stability, demonstrating their suitability for use in complex biological environments (Supplementary Fig. 35). In a proof-of-concept experiment, we dispersed 1,2-distearoyl-sn-glycero-3-phosphoethanolamine-N-[methoxy(polyethylene glycol)-2000] (DSPE-mPEG(2000))-modified NMF-H-0 and NMF-H-3.1 NCs (separately) into 0.9 wt% normal saline at a safe dosage (200 μL, 20 mg/mL, Supplementary Fig. 36), then intravenously injected the prepared solutions into living nude mice. Real-time vascular imaging was performed in vivo with a custom NIR-II microscopic bioimaging system using 980 nm diode laser excitation (power density: ~0.10 W/cm², Fig. 5f). Benefiting from the significantly enhanced NIR-II DCL of Er³⁺, the whole-body vascular networks of the living nude mice, including the small capillary blood vessels branching from the

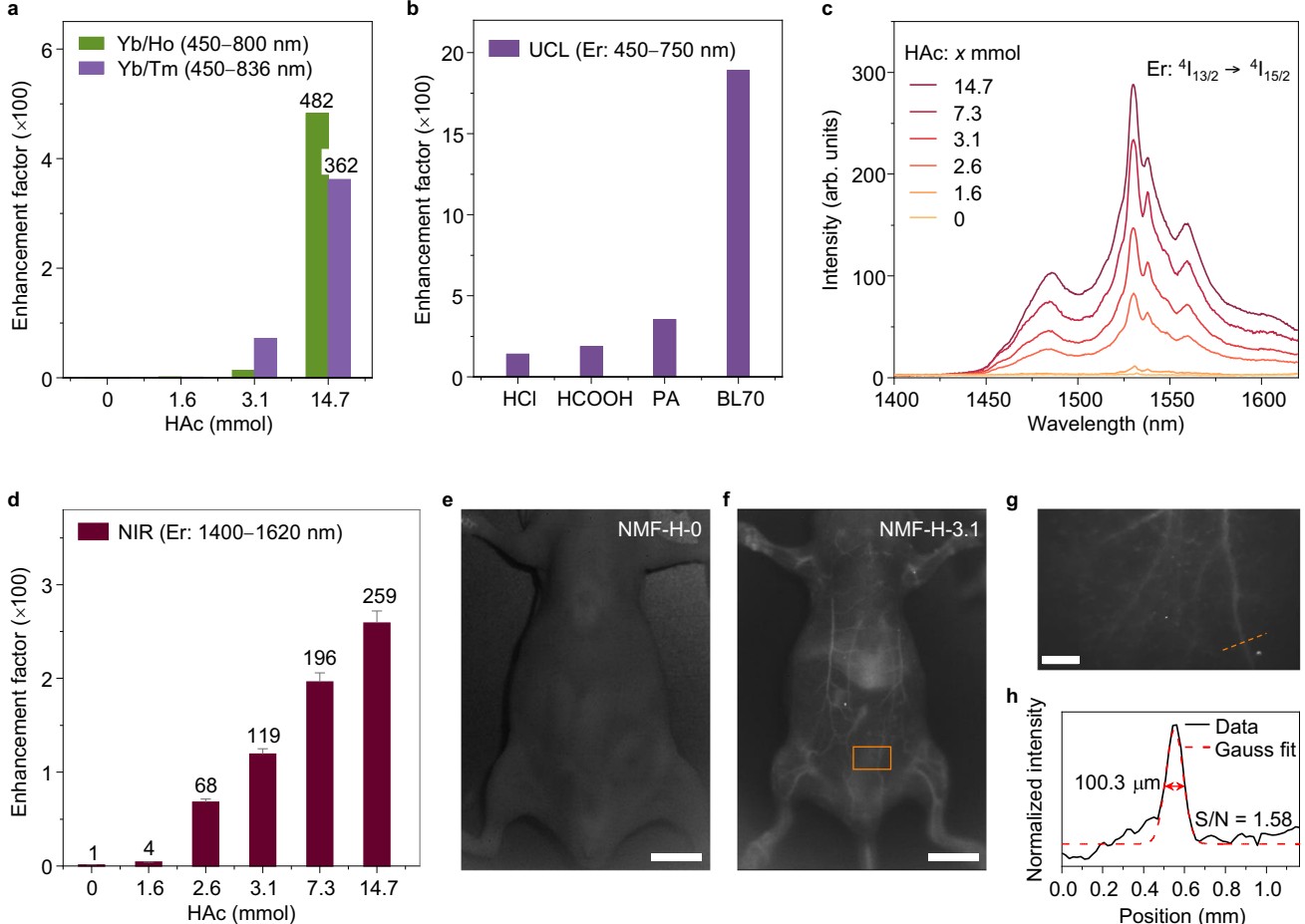

**Fig. 5 | Generalizability of interstitial H⁺-doping strategy and demonstration of in vivo imaging via downconversion luminescence (DCL) second near-infrared (NIR-II) emission. a** Luminescence enhancement factor of Yb/Tm- and Yb/Ho-co-doped $NaMgF_3$ nanocrystals (NCs) with different nominal acetic acid (HAc) additions. Green bars: $NaMgF_3$:Yb/Tm; purple bars: $NaMgF_3$:Yb/Ho. **b** UCL enhancement factors of $NaMgF_3$:Yb/Er upconversion NCs synthesized with hydrochloric acid (HCl), formic acid (HCOOH), propionic acid (PA), or benzenesulfonic acid (BL70; $PhSO_3H$) as the H⁺ precursor. **c** Typical DCL NIR-II emission spectra of $NaMgF_3$:Yb/Er (NMF-H-X) NCs synthesized with different nominal amounts of HAc (0–14.7 mmol), and **d** corresponding emission enhancement factors. The error bars in **d** represent the standard deviation of the luminescence enhancement factor.

**e, f** In vivo NIR-II images of BALB/c nude mice after tail-vein injection of a dispersion of 1,2-distearoyl-sn-glycero-3-phosphoethanolamine-N-[methoxy(polyethylene glycol)-2000] (DSPE-mPEG(2000))-modified NMF-H-0 (**e**) or NMF-H-3.1 (**f**) NCs in 0.9 wt% normal saline at a safe dosage (200 μL, 20 mg/mL). Scale bar: 10 mm. **g** Vascular fluorescence image of the abdomen of a representative BALB/c nude mouse (region of interest: orange box in **f**) as displayed in the NIR-II 1532 nm window (1300 nm long-pass filter, exposure time 400 ms). Scale bar: 1 mm. **h** Analysis of vessel FWHM and signal-to-noise ratio (S/N), based on the cross-sectional intensity (data and Gaussian fit) profiles along the orange dashed line in **g**. Source data are provided as a Source Data file.

main arteries, were clearly delineated by the NIR-II DCL signal of $Er^{3+}$ when utilizing NMF-H-3.1 NCs (mean crystal size: 12.3 ± 1.1 nm) as a bioimaging agent (Fig. 5f, g). This result starkly contrasted that with NMF-H-0 NCs, in which hardly any detectable NIR-II signals were observed under otherwise identical conditions (Fig. 5e). A cross-sectional intensity profile of a randomly selected capillary blood vessel in the abdomen of a nude mouse delivered a Gaussian-fitted full-width at half-maximum (FWHM) of as little as ~100.3 μm and a high signal-to-noise ratio of 1.58 (Fig. 5h). Although preliminary, these vascular mapping results confirm that interstitially H⁺-doped $NaMgF_3$:Yb/Er NCs are promising bioimaging agents for high-contrast cardiovascular bioimaging.

## Discussion

In conclusion, we directly observed the crystal-field-perturbation-enhanced luminescence of $Ln^{3+}$-doped NCs using a simple but effective interstitial H⁺-doping strategy. Both the UCL and NIR-II DCL of the $Er^{3+}$ emitters in orthorhombic $NaMgF_3$:Yb/Er NCs were enhanced without increasing the crystal size. The experimental results and first-principles DFT calculations confirmed that interstitial H⁺-doping exerts a crystal-field perturbation effect by inducing the anisotropic polarization of the ligand (F⁻), which partially mitigates the fundamental constraint of parity-forbidden $4f$–$4f$ transitions of $Er^{3+}$ activators and increases the radiative transition probability ($A_{ed}$) of the excited state and excitation light absorption ability of the ground state, thereby achieving significant $Er^{3+}$ UCL enhancement (by up to 1891-fold in our experiments). This fundamental understanding provides deeper insight into the use of $4f$ electrons for constructing small NCs with high-brightness $Ln^{3+}$ emission for diverse optical imaging and biomedical applications.

## Methods

### Materials

$Mg(CH_3CO_2)_2 \cdot 4H_2O$ (99%), $Ca(CH_3CO_2)_2 \cdot 4H_2O$ (99%), $Y(CH_3CO_2)_3 \cdot 4H_2O$ (99.99%), $Yb(CH_3CO_2)_3 \cdot 4H_2O$ (99.999%), $Er(CH_3CO_2)_3 \cdot 4H_2O$ (99.99%), $Ho(CH_3CO_2)_3 \cdot 4H_2O$ (99.99%), $Tm(CH_3CO_2)_3 \cdot 4H_2O$ (99.99%), HAc (≥99.7%), NaAc (≥99.0%), BL70 (98%), oleic acid (OA, technical grade 90%), and trioctylamine (TOA, 98%) were purchased from

Sigma-Aldrich (China). NaOH (96%), ammonium fluoride ($NH_4F$, 98%), HCOOH (99%), and chloroform ($CHCl_3$) were purchased from Aladdin (China). Sodium chloride (NaCl), sodium formate (HCOONa), sodium benzenesulfonate ($PhSO_3Na$), and PA were purchased from Adamas-Beta (China). Cyclohexane ($C_6H_{12}$), methanol ($CH_3OH$), HCl, and ethanol ($C_2H_6O$) were purchased from Sinopharm Chemical Reagent Co. (China). DSPE-mPEG(2000) was purchased from Xi'an Ruixi Biological Technology Co., Ltd. (China). Cell Counting Kit-8 (CCK-8), human kidney epithelial (293T) cells, BALB/c nude mice, Dulbecco's Modified Eagle Medium, fetal bovine serum, and penicillin–streptomycin were purchased from Beijing Dingguo Changsheng Biotechnology Co., Ltd. (China). Deionized water was used wherever water was required. All chemicals were used as received without further purification.

## Preparation of NaMgF₃:Yb/Er NCs without interstitial H⁺-doping (NMF-H-0 NCs)

NMF-H-0 NCs were synthesized via a modified high-temperature coprecipitation method[2]. In a typical procedure, 0.475 mmol $Mg(CH_3CO_2)_2 \cdot 4H_2O$, 0.02 mmol $Yb(CH_3CO_2)_3 \cdot 4H_2O$, 0.005 mmol $Er(CH_3CO_2)_3 \cdot 4H_2O$, 5 mL OA, and 15 mL TOA were added to a 100 mL three-neck flask and degassed under $N_2$ flow at 25 °C for 20 min. The solution was heated to 155 °C and kept at this temperature under $N_2$ flow with constant stirring for 30 min to form a clear solution, and then cooled naturally to 50 °C. Thereafter, 10 mL of a methanol solution containing 1.5 mmol NaOH and 1.5 mmol $NH_4F$ was added, and the resulting mixture was heated to 75 °C and stirred for 30 min under 3 L/min $N_2$ flow. The methanol was considered to have been completely removed by evaporation when the solution no longer produced bubbles. Then, the resultant solution was heated to 310 °C under $N_2$ flow with vigorous stirring for 60 min, and then cooled naturally to 50 °C. The resulting NMF-H-0 NCs were precipitated by the addition of 30 mL acetone and collected by centrifugation at $16,710 \times g$ for 5 min. The precipitates were redispersed in 5 mL cyclohexane solution by ultrasonication, then mixed with 15 mL ethanol and collected by centrifugation at $16,710 \times g$ for 5 min. After repeating the above operation twice, the final product was redispersed and stored in 5 mL cyclohexane.

NMF-H-0 NCs with different $Ln^{3+}$ ions dopants were prepared by replacing the $Er(CH_3CO_2)_3 \cdot 4H_2O$ addition with $Ho(CH_3CO_2)_3 \cdot 4H_2O$ or $Tm(CH_3CO_2)_3 \cdot 4H_2O$. To prepare NMF-H-0 NCs with $Ac^-$ ions, the $Na^+$ source (NaOH) was replaced with NaAc. To investigate the effect of different conjugate bases, NMF-H-0 NCs were prepared in which the $Na^+$ source (NaOH) was replaced with NaAc, NaCl, HCOONa, or $PhSO_3Na$. All other reagents, conditions, and quantities were kept constant.

## Preparation of NaMgF₃:Yb/Er NCs with interstitial H⁺-doping (NMF-H-X NCs)

The experimental setup used to achieve interstitial $H^+$ doping is illustrated in Supplementary Fig. 4. The synthesis method was essentially the same as that for the preparation of NMF-H-0 NCs, except that the solution was cooled naturally to 50 °C after the complete removal of methanol. Subsequently, nominal amounts of HAc (X = 1.6, 2.6, 3.1, 7.3, 14.7, 20, and 25 mmol) were added to the three-neck flask. The solution was rapidly heated to 310 °C, stirred vigorously for 60 min, and cooled naturally to 50 °C. To prevent a decrease in the HAc content owing to evaporation, the $N_2$ flow rate was set to 0.1 L/min (Supplementary Fig. 5). The resultant NMF-H-X NCs were precipitated by the addition of 30 mL acetone and collected by centrifugation at $16,710 \times g$ for 5 min. The precipitates were redispersed in 5 mL cyclohexane solution by ultrasonication, then mixed with 15 mL ethanol and collected by centrifugation at $16,710 \times g$ for 5 min. After repeating the above operation twice, the final product was redispersed and stored in 5 mL cyclohexane.

NMF-H-X NCs with different $Ln^{3+}$ ions dopants were prepared by replacing the $Er(CH_3CO_2)_3 \cdot 4H_2O$ addition with $Ho(CH_3CO_2)_3 \cdot 4H_2O$ or $Tm(CH_3CO_2)_3 \cdot 4H_2O$, and NMF-H-X NCs with different $H^+$ sources were prepared by replacing the HAc precursor with HCl, HCOOH, PA, or BL70. All other reagents, conditions, and quantities were kept constant.

## Preparation of NaMgF₃ inert shell precursor

In a typical procedure, 0.5 mmol $Mg(CH_3CO_2)_2 \cdot 4H_2O$, 2.5 mL OA, and 7.5 mL TOA were added to a 50 mL three-neck flask and degassed under $N_2$ flow at room-temperature for 20 min. The solution was heated to 155 °C and kept at this temperature under $N_2$ flow with constant stirring for 30 min to form a clear solution, followed by cooling to 50 °C. Thereafter, 10 mL of a methanol solution containing 1.5 mmol NaOH and 1.5 mmol $NH_4F$ was added, and the resulting mixture was heated to 75 °C and stirred for 30 min under 3 L/min $N_2$ flow. Upon the complete removal of the methanol by evaporation (i.e., when the obtained solution no longer produced bubbles), the $NaMgF_3$ inert shell precursor solution was obtained.

## Preparation of NMF-H-0/NMF-H-14.7@NaMgF₃ core–shell NCs

NMF-H-0@NaMgF₃ and NMF-H-14.7@NaMgF₃ core–shell NCs were synthesized by similar methods. Here, the synthesis of NMF-H-0@NaMgF₃ NCs is described as a typical example. A 2.5 mL cyclohexane solution of NMF-H-0 NCs (20 mg/mL) was mixed with 2.5 mL OA and 7.5 mL TOA in a 100 mL three-neck flask and degassed under $N_2$ flow at 25 °C for 20 min. Then, the mixed solution was heated to 110 °C to remove cyclohexane and water. Subsequently, the mixed solution was heated to 310 °C at a rate of 20 °C/min under a $N_2$ atmosphere. Thereafter, 10 mL of the $NaMgF_3$ inert shell precursor solution was slowly injected into the three-neck flask using a syringe pump at a rate of 300 mL/min. After the injection of the $NaMgF_3$ inert shell precursor, the reaction system was held at 310 °C for another 30 min and then allowed to cool naturally to 50 °C. The thickness of the inert $NaMgF_3$ shell layer was readily tuned by adjusting the amount of $NaMgF_3$ inert shell precursor and injection rate. The resultant $NaMgF_3$:Yb/Er@NaMgF₃ core–shell NCs were precipitated by the addition of 30 mL acetone and collected by centrifugation at $16,710 \times g$ for 5 min. The precipitates were redispersed in 5 mL cyclohexane solution by ultrasonication, then mixed with 15 mL ethanol and collected by centrifugation at $16,710 \times g$ for 5 min. After repeating the above operation twice, the final product was redispersed and stored in 5 mL cyclohexane.

The experimental steps for the synthesis of the NMF-H-14.7@NaMgF₃ NCs were the same as those for the synthesis of the NMF-H-0@NaMgF₃ NCs, except that the 2.5 mL NMF-H-0 NC cyclohexane solution was replaced with 2.5 mL of NMF-H-14.7 NC cyclohexane solution.

## Preparation of NMF-H-7.3@NaMgF₃:H core–shell NCs

The NMF-H-7.3@NaMgF₃:H core–shell NCs were synthesized by a similar method to that used for the NMF-H-0/NMF-H-14.7@NaMgF₃ core–shell NCs[12]. A 2.5 mL cyclohexane solution of NMF-H-7.3 NCs (20 mg/mL) was precipitated, centrifuged with ethanol, and redispersed in OA to a total volume of 2.5 mL. Next, the oleic acid solution of NMF-H-7.3 NCs was mixed with TOA (7.5 mL) in a 100 mL three-neck flask and degassed under $N_2$ flow at 25 °C for 20 min. The mixed solution was then heated to 110 °C to remove the residual cyclohexane and ethanol. Thereafter, 10 mL of the $NaMgF_3$ inert shell precursor solution mixed with 7.3 mmol HAc was added directly into the three-neck flask. Subsequently, the mixed solution was heated to 310 °C at a rate of 20 °C/min under a $N_2$ atmosphere, maintained for 60 min, and cooled naturally to 50 °C. The resultant NMF-H-7.3@NaMgF₃:H core–shell NCs were precipitated by the addition of 30 mL acetone and collected by centrifugation at $16,710 \times g$ for 5 min. The precipitates

were then redispersed in 5 mL cyclohexane solution by ultrasonication, mixed with 15 mL ethanol, and collected by centrifugation at $16,710 \times g$ for 5 min. After repeating the above operation twice, the final product was redispersed and stored in 5 mL cyclohexane.

### Preparation of CaF$_2$:Yb/Er NCs without/with interstitial H$^+$-doping (CaF-H-X NCs)

The synthesis of the CaF-H-0 NCs was similar to that of the NMF-H-0 NCs. In a typical procedure, 0.39 mmol Ca(CH$_3$CO$_2$)$_2$·4H$_2$O, 0.10 mmol Yb(CH$_3$CO$_2$)$_3$·4H$_2$O, 0.01 mmol Er(CH$_3$CO$_2$)$_3$·4H$_2$O, 5 mL OA, and 15 mL TOA were added to a 100 mL three-neck flask and degassed under N$_2$ flow at 25 °C for 20 min. The solution was heated to 155 °C and kept at this temperature under N$_2$ flow with constant stirring for 30 min to form a clear solution, and then cooled naturally to 50 °C. Thereafter, 10 mL of a methanol solution containing 1 mmol NaOH and 1.5 mmol NH$_4$F was added, and the resulting mixture was heated to 75 °C and stirred for 30 min under 3 L/min N$_2$ flow. The methanol was considered to have been completely removed by evaporation when the solution no longer produced bubbles. Then, the resultant solution was heated to 280 °C under N$_2$ flow with vigorous stirring for 60 min, and cooled naturally to 50 °C. The resulting CaF-H-0 NCs were precipitated by adding 30 mL acetone and collected by centrifugation at $16,710 \times g$ for 5 min. The precipitates were redispersed in 5 mL cyclohexane solution by ultrasonication, then mixed with 15 mL ethanol and collected by centrifugation at $16,710 \times g$ for 5 min. After repeating the above operation twice, the final product was redispersed and stored in 5 mL cyclohexane.

The synthesis of interstitially H$^+$-doped CaF$_2$:Yb/Er NCs was similar to that of the interstitially H$^+$-doped NaMgF$_3$:Yb/Er NCs, with nominal HAc additions of $X = 0$, 3.1, 7.3 mmol. All other reagents, conditions, and quantities were kept constant.

### Preparation of NaYF$_4$:Yb/Er NCs without/with interstitial H$^+$-doping (NYF-H-X NCs)

The synthesis of the NYF-H-0 NCs was similar to that of the NMF-H-0 NCs. In a typical procedure, 0.39 mmol Y(CH$_3$CO$_2$)$_3$·4H$_2$O, 0.10 mmol Yb(CH$_3$CO$_2$)$_3$·4H$_2$O, 0.01 mmol Er(CH$_3$CO$_2$)$_3$·4H$_2$O, 5 mL OA, and 15 mL TOA were added to a 100 mL three-neck flask and degassed under N$_2$ flow at 25 °C for 20 min. The solution was heated to 155 °C and maintained at this temperature under N$_2$ flow with constant stirring for 30 min to form a clear solution, and cooled naturally to 50 °C. Thereafter, 10 mL of a methanol solution containing 1.25 mmol NaOH and 2 mmol NH$_4$F was added, and the resulting mixture was heated to 75 °C and stirred for 30 min under 3 L/min N$_2$ flow. The methanol was considered to have been completely removed by evaporation when the solution no longer produced bubbles. Then, the resultant solution was heated to 300 °C under N$_2$ flow with vigorous stirring for 60 min, and cooled naturally to 50 °C. The resulting NYF-H-0 NCs were precipitated by adding 30 mL acetone and collected by centrifugation at $16,710 \times g$ for 5 min. The precipitates were redispersed in 5 mL cyclohexane solution by ultrasonication, then mixed with 15 mL ethanol and collected by centrifugation at $16,710 \times g$ for 5 min. The final product was redispersed and stored in 5 mL cyclohexane.

The synthesis of interstitially H$^+$-doped NaYF$_4$:Yb/Er NCs was similar to that of the interstitially H$^+$-doped NaMgF$_3$:Yb/Er NCs, with nominal HAc additions of $X = 0$–19.2 mmol. All other reagents, conditions, and quantities were kept constant.

### General procedure for the synthesis of ligand-free NaMgF$_3$:Yb/Er NCs

Ligand-free NaMgF$_3$:Yb/Er NCs were obtained by removing the surface ligands of the oleate-capped counterparts through acid treatment[47]. In a typical process, approximately 15 mg of the as-synthesized oleate-capped NaMgF$_3$:Yb/Er NCs were dispersed in 15.08 mL of acidic ethanol solution (prepared by adding 80 µL of concentrated HCl to 15 mL

of absolute ethanol) and ultrasonicated for 20 min to remove the surface ligands. After the reaction, the NCs were collected by centrifugation at $16,710 \times g$ for 10 min, and further purified using an acidic ethanol solution (prepared by adding 5 µL of concentrated HCl to 15 mL of absolute ethanol). The resulting products were washed with 15 mL ethanol, dried in an oven at 70 °C for 6 h, and ground into powder. The solid powders were freeze-dried for 2 days to remove residual water from the surface of the NaMgF$_3$:Yb/Er NCs and stored in an Ar atmosphere at 4 °C for subsequent SSNMR characterization.

### Surface modification of NMF-H-0 and NMF-H-3.1 NCs with DSPE-mPEG(2000)

The surface of the NMF-H-0 and NMF-H-3.1 NCs was modified using DSPE-mPEG(2000) to enhance the water solubility and biocompatibility[48]. A cyclohexane dispersion of NMF-H-0 or NMF-H-3.1 NCs (1 mL, 20 mg/mL) was added to 5 mL ethanol, collected via centrifugation at $16,710 \times g$ for 5 min, and then redispersed in 4 mL of chloroform. In addition, 100 mg of DSPE-mPEG(2000) was dissolved in 6 mL of chloroform. Then, the abovementioned solutions were mixed and stirred overnight, following which a rotary evaporator was used to evaporate the chloroform for approximately 1 h at 60 °C and 0.1 Pa. The residue was redispersed in 4 mL of deionized water, after which the DSPE-mPEG(2000)-modified NCs were centrifuged at $27,579 \times g$ for 30 min to remove free DSPE-mPEG(2000) and chloroform. The residue was finally ultrasonically dispersed in 0.9 wt% normal saline to form a clear DSPE-mPEG(2000)-coated NC solution with a concentration of 20 mg/mL.

### Characterization

Powder XRD patterns of all the NCs were collected using an X-ray diffractometer (MiniFlex2, Rigaku) with Cu Kα radiation ($\lambda = 0.154187$ nm) in the $2\theta$ range of 20°–75° at a scanning rate of 0.2°/min. Transmission electron microscopy (TEM) images were acquired on a TECNAI G2F20 TEM. High-resolution TEM, HAADF-STEM, and two-dimensional EDS elemental mapping of the as-synthesized NMF-H-X NCs were performed using a Titan G2 80–200 Chemi STEM FEI TEM. X-ray photoelectron spectroscopy (XPS) was conducted on a Thermo Fisher ESCALAB 250Xi spectrometer using Al Kα (1486.6 eV) radiation. The actual dopant concentration of the Yb/Er cations was determined using an iCAP7400 ICP-OES spectrometer. UCL spectra were measured on a spectrometer equipped with both continuous (450 W) xenon pulsed flash lamps and a 980 nm diode laser (FLS920, Edinburgh Instrument). The absolute UCQY of the samples was measured with a custom UCL spectroscopy system at room-temperature upon 980 nm diode laser excitation at a power density of 50 W/cm$^2$, and the UCL peaks from the Er$^{3+}$ ions in the spectral range of 400–750 nm were integrated for the UCQY determination. NIR-II luminescence spectra were measured with a FLS980 fluorescence spectrometer (Edinburgh Instrument) equipped with continuous xenon arc lamp (450 W). The absolute NIR quantum yields (NIR-QYs) of the as-synthesized NMF-H-X NCs were measured in a barium sulfate-coated integrating sphere (Edinburgh) mounted on a FLS980 spectrometer, with the input and output ports of the sphere located at 90° from each other in the plane of the spectrometer. Red UCL digital photographs of the NMF-H-X NCs were taken with a Canon EOS 5D Mark IV camera upon 980 nm continuous-wave laser excitation without any filter. Fourier-transform infrared (FTIR) spectra were recorded on a Magna 750 FTIR spectrometer. $^{19}$F-SSNMR spectra were collected using a single pulse method. Experiments were performed on a Bruker AVANCE III-500 WB spectrometer (Bruker BioSpin) equipped with a 4 mm magic-angle spinning (MAS) probe. The MAS spinning frequency, pulse width, and recycle delay time were 14 kHz, 3.4 s, and 8 s, respectively. The absorption spectra were measured on a Perkin Elmer UV-VIS-NIR Lambda 950 double beam spectrophotometer. NIR-II bioimaging images of nude mice were taken using

an in vivo fluorescence imaging system (NirVivo-MIX, RayLight). EXAFS measurements were carried out using the XAS Beamline at the Australian Synchrotron (ANSTO) in Melbourne, Australia, using a set of liquid-nitrogen-cooled Si (111) monochromator crystals. The electron beam energy was 3.0 GeV. With the associated beamline optics (Si-coated collimating mirror and Rh-coated focusing mirror), the harmonic content of the incident X-ray beam was negligible. A Ge (100) detector was used to collect the fluorescence signals, and the energy was calibrated using a Yb foil. The beam spot size was approximately $1 \times 1$ mm. Each XAS scan required approximately 1 h. EXAFS fitting was conducted in Athena and Artemis software[49]. First-principles calculations were performed at the LvLiang Cloud Computing Center, China; the Supercomputer Center of Fujian Institute of Research on the Structure of Matter (FJIRSM), China; and Shenzhen Huashan Technology Co., Ltd., China. UCL decays were measured by using a tunable mid-band optical parametric oscillator (OPO) pulse laser as the excitation source (410–2400 nm, 10 Hz, pulse width ≤5 ns, Vibrant 355II, OPOTEK). The effective lifetime ($\tau_{\text{eff}}$) was determined by[12]

$$\tau_{\text{eff}} = \frac{1}{I_0} \int_0^\infty I(t)dt, \qquad (1)$$

where $I_0$ and $I(t)$ represent the maximum luminescence intensity and luminescence intensity at time $t$ after the excitation light was cut off, respectively. Each experiment was repeated at least three times to ensure the accuracy of the experimental results.

## Cell culture and in vitro biotoxicity of DSPE-mPEG(2000)-modified NMF-H-X NCs

The in vitro biotoxicity of the DSPE-mPEG(2000)-modified NMF-H-X NCs was tested by standard CCK-8 cytotoxicity assays on human kidney epithelial (293 T) cells[50]. In brief, 293 T cells were seeded in a 96-well plate at $1 \times 10^4$ cells/well and cultured in Dulbecco's Modified Eagle Medium with 10% fetal bovine serum and 1% penicillin–streptomycin (37 °C, 5% $CO_2$) for 24 h. Thereafter, different concentrations of NMF-H-X NCs (0, 0.25, 0.5, 1, 5, 10, 50, 100, 200, and 500 µg/mL, diluted in 0.9 wt% normal saline) with three parallel tests were added to the wells. After incubating with the NCs at 37 °C under 5% $CO_2$ for 48 h, CCK-8 solution (10 µL) was added to each well and the plate was incubated for an additional 2 h at 37 °C under 5% $CO_2$. The absorbances at 450 nm of each well were measured using a multimodal microplate reader (Synergy 4, BioTek). The inhibition of cell growth was evaluated by calculating the percentage cell viability as the ratio of the mean of absorbance values of the treatment group to the control group. The viability of the control sample without NMF-H-X NCs was set as 100%.

## In vivo NIR-II fluorescence imaging

The luminescent signals from NMF-H-3.1 and NMF-H-0 NCs were collected using an InGaAs camera (Ninox 640 SU, Raptor Photonics, UK) equipped with a 1300 long-pass filter for NIR-II fluorescence imaging. The excitation light was provided by a 980 nm continuous-wave laser (Laserwave, China). Nude mice (7–8 weeks old, specific pathogen free (SPF)) were tail-vein injected with DSPE-mPEG(2000)-modified NMF-H-3.1 or NMF-H-0 NCs in 0.9 wt% normal saline (20 mg/mL, 200 µL) under anesthesia. At 10 min post-injection, whole-body and abdomen vessel imaging was conducted using the NIR-II in vivo imaging system. To ensure the quality of the images, the excitation power density of the laser was set at 100 mW/cm² and the exposure time of the camera was set at 400 ms. Animal studies were approved by the Fujian Medical University of Ethical Committees on Experimental Animal Care and Use (IACUC FJMU 2023-0007) and performed in accordance with institutional and national guidelines.

## Computational details

To gain deeper insight into the mechanism by which the insertion of $H^+$ ions enhances the luminescence of pure and/or $Ln^{3+}$-doped $NaMgF_3$ NCs, first-principles DFT calculations were performed using the Vienna Ab initio Simulation Package (VASP)[51,52]. The generalized gradient approximation (GGA), Perdew–Burke–Ernzerhof (PBE) exchange–correlation functional, and projector augmented wave (PAW) method were used for all calculations[51,52]. The generalized criteria for the residual force and energy during structural relaxation were set as 0.03 eV/Å and $10^{-5}$ eV, respectively. The plane-wave cutoff energy was set as 520 eV, with $2 \times 3 \times 2$ $k$-point sampling of the Brillouin zone. A $2 \times 2 \times 1$ supercell of $NaMgF_3$ was selected as the basic structure, which was first fully relaxed and then modified to construct models with several possible point defects, such as interstitial $H^+$, substitutional $Yb^{3+}$, substitutional $H^+$, and $Na^+$ vacancies, as well as combinations of these defects. This enabled us to reflect on the possible structures in the real crystal. For example, replacing $Mg^{2+}$ with $Yb^{3+}$ would lead to a positively charged system in a perfect supercell, or a neutral system if a $Na^+$ vacancy was also present. In this manner, several of the possible structures that could be constructed by inserting isolated $H^+$ ions were simulated. For these simulated systems, their structures were fully relaxed to the energetic minima for subsequent calculations, and the NELECT parameter was adjusted to account for charge effects.

The formation energy ($\Delta E_{\text{form}}$) in charge state $q$ was defined as the energy difference between the NC and its isolated atoms[51,53]. This is usually considered a reliable indicator for material stability and is associated with the chemical potential of the atoms and electrons[51,53]. Thus, the $\Delta E_{\text{form}}$ values were calculated for the pure $NaMgF_3$ and $NaMgF_3$:Yb/H NCs with possible point defects, such as substitutional $Yb^{3+}$ additions on $Mg^{2+}$ sites ($Yb^{3+}_{Mg}$), $Na^+$ vacancies ($V_{Na}$), interstitial $H^+$ additions ($H_i$), and substitutional $H^+$ additions on $Na^+$ sites ($H^+_{Na}$). The $\Delta E_{\text{form}}$ values for the $NaMgF_3$ supercells with and without defects were calculated by using the following simplified equation[51–53]: $\Delta E_{\text{form}} = E_{\text{total(defect)}} - E_{\text{total(perfect)}} - \Sigma \eta_i \mu_i + q(E_F + E_{VBM})$, where $E_{\text{total(defect)}}$ and $E_{\text{total(perfect)}}$ are the total energies of the defective and perfect supercells, respectively; $\eta_i$ is the number of atoms of type $i$ that have been added to ($\eta_i > 0$) or removed from ($\eta_i < 0$) the perfect supercell; $\mu_i$ is the corresponding chemical potential; $q$ is the number of electrons transferred from the supercell to the Fermi reservoirs when forming the defective supercell; and $E_F$ and $E_{VBM}$ are the energies of the Fermi level and valence-band maximum (VBM), respectively, in the perfect supercell. The band structure and density of states were determined by DFT calculation based on the former optimized structures. In addition, the band structures, density of states, electron localization function, and transition dipole moment were elaborated and analyzed with the aid of the VASPKIT software[54].

## Data availability

All data supporting the findings of this study are present in this paper and its Supplementary Information. Source data are available from the corresponding authors upon request. Source data are provided with this paper.

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

## Acknowledgements

This work was supported by the Fund of Fujian Science & Technology Innovation Laboratory for Optoelectronic Information (2020ZZ114 and 2022ZZ204, Y.L.), Key Research Program of Frontier Science CAS (QYZDY-SSW-SLH025, M.H.), Fund of Advanced Energy Science and Technology Guangdong Laboratory (DJLTN0200/DJLTN0240, M.H.), and Natural Science Foundation of Fujian Province (No. 2022J05102, J.K.). The authors thank Dr. Liangliang Liang, Mr. Yuanchao Lei, Dr. Xiao Qin, Mr. Peng Li, and Mr. Yingjie Zhao for their contributions to the literature review and graphics suggestion.

## Author contributions

G.L., Y.L., and M.H. conceived, designed, and supervised the project and led the collaboration efforts. G.L. and S.J. synthesized the samples. G.L. performed the characterization and optical measurements with contributions from S.J., A.L., and J.K. Quantum mechanical calculations were conducted by C.L. Biological experiments were performed by L.Y. and S.J. The manuscript was written by G.L., Y.L., and L.C. All authors participated in the discussion and analysis of the manuscript.

## Competing interests

The authors declare no competing interests.
