## [Peer Review File · Nature Communications]

REVIEWER COMMENTS

Reviewer #1 (Remarks to the Author):

The manuscript is focused on demonstrating an exciting strategy to improve the emission amplification of Lanthanide ions doped nanocrystals by inserting interstitial H⁺ through the computational and experimental procedure. The hypothesis presented by the authors is that the crystal-field perturbation breaks the parity-forbidden nature of 4f-transitions of trivalent lanthanide by inserting interstitial H⁺, leading to a significant emission amplification. The work provides a relevant contribution to designing new materials with desired characteristics for several applications. The detailed discussion of the data obtained led the study to well-founded conclusions. I recommend that the article be accepted after some doubts are cleared.

1) How electronic states of impurities are deposited in the band gap?

2) The J-mixing effect in Ln³⁺ ions created by a distortion in local symmetry is one of the main factors that violate the Laporte rule. But the results presented by the authors don't show large distortions in the local symmetry of the lanthanide. What kind of effect is happening for this perturbation of the crystal field to occur? In my humble opinion, the entry of hydrogen may be favoring dynamic coupling. In this case, oscillating dipoles induced on the surrounding produce a field that changes the transition dipole moment of the emitter. See for example the following papers: <https://doi.org/10.1103/PhysRevLett.81.1381>; <https://doi.org/10.1016/j.jlumin.2021.118292>; [https://doi.org/10.1016/0022-3697\(95\)00020-8](https://doi.org/10.1016/0022-3697(95)00020-8); [https://doi.org/10.1016/0022-5088\(83\)90455-1](https://doi.org/10.1016/0022-5088(83)90455-1);

3) Did the authors verify the defect clustering with the lanthanide entering into the Na site by DFT calculations? What is the defect formation energy?

Reviewer #2 (Remarks to the Author):

This manuscript reports the synthesis of trivalent lanthanide-based nanocrystals (NCs) with the formula NaMgF₃:Ln³⁺ or NMF-H-X, where X is the acid concentration (mmol), Ln³⁺ = Yb³⁺ and Er³⁺, and NaMgF₃ crystals are used as a model host matrix. Due to parity-forbidden 4f-4f transitions, these NCs exhibit weak upconversion (UC) and downconversion (DC) luminescence. The authors found that the interstitial H⁺-doping with acetic acid, formic acid, hydrochloric acid, and benzenesulfonic acid (BL70) can drastically improve the UC and DC luminescence intensities of these NCs. After using BL70 as a proton source, NMF-H-0.5 displayed an increase in UC luminescence intensity of 1890.6-fold. The authors hypothesized that crystal-field perturbation, which aids in breaking the parity-forbidden nature of 4f-transitions in these NCs, causes this sharp increase in UC luminescence intensities after H⁺-doping. Finally, H⁺-doped NaMgF₃:Yb/Er NCs with NIR-II emission was used to perform high-resolution angiography on nude mice. Overall, the work seems good, but several unanswered questions must be explained. I recommend that the manuscript be accepted by "Nature Communications" after major revision. The detailed comments are shown below.

(1) Do the conjugate bases, such as chloride ion (Cl⁻), formate ion (HCOO⁻), or benzenesulfonate (PhSO₃⁻), affect the size of the NCs? Can conjugate base PhSO₃⁻ affect the parity-forbidden 4f transitions?

(2) What makes the shapes of octahedra and dodecahedra appropriate for interstitial H⁺-doping?

(3) In the case of BL70, why is the UC luminescence intensity of NMF-H-0.5 found to be higher than that of NMF-H-0.8?

(4) Why does the addition of 1.6 mmol of acetic acid decrease the crystal size from 12.6 Å to 6.7 Å, but the addition of 3.1 mmol of acetic acid increases the crystal size to 12.3 Å?

(5) Why does the UC emission intensity of the 4S_{3/2}→4I_{15/2} transition have no effect when using acetic acid as a proton source but changes in the presence of HCl?

(6) Comparing the UC luminescence intensities of NMF-H-0, NMF-H-14.7, NMF-H-0 (with NaOAc), and NMF-H-0 (with NaOH) is recommended.

(7) Why is the UCL intensity of NMF-H-0 (with NaOAc) higher than NMF-H-0 (with NaOH)?

(8) What role does NaOH play in synthesizing NMF-H-0 NC? Can NaOAc be used instead?

(9) The quantum yield of all samples should be measured and reported in the manuscript.

(10) Can temperature affect the 4F_{9/2}→4I_{15/2} transition?

(11) Why did the authors choose acetic acid, formic acid, hydrochloric acid, and benzenesulfonic acid (BL70) as proton sources?

(12) The authors are encouraged to change the word "proof" to "report" in the "Abstract" section.

(13) The phrase "see Methods online" is used frequently by the authors. In this regard, kindly provide the appropriate references.

Reviewer #3 (Remarks to the Author):

This paper entitled "Proof of Crystal-Field-Perturbation-Enhanced Luminescence of Lanthanide-Doped Nanocrystals through Interstitial H⁺ doping" reports a new crystal engineering approach for enhancing the luminescence of lanthanide-doped upconversion nanoparticles. By alleviating the parity-forbidden selective rule of Ln³⁺ 4f-4f radiative transition by interstitial H⁺-doping, the authors achieved a three orders of magnitude emission enhancement in the orthorhombic phase NaMgF₃: Ln³⁺. Although the results are interesting, some problems should be first addressed to fulfil the publication requirements for Nature Communications, referring to my comments below.

1. Under high temperature and vacuum conditions, acetic acid exhibits relatively high volatility. To provide more clarity on the preparation procedure, the authors should specify parameters such as the vacuum time and temperature. It is recommended that a control experiment should be conducted to determine the effect of varying vacuum time and temperature on H⁺ doping.
2. Although the orthorhombic NaMgF₃ has been chosen as a model host for demonstration, the relatively low rare earth doping concentration (less than 10%) and low emission efficiency diminish the persuasiveness of the results. It is important to investigate whether this strategy can be applied to other, more efficient host matrices, such as CaF₂, cubic-NaYF₄ and hexagonal NaYF₄.
3. Due to the small ionic radius of H⁺, it is uncertain whether it will remain stable in the interstitial site, particularly during ultrasonic dispersion. Moreover, if it is stable, the long-term stability (more than one month) should be demonstrated.
4. As the amount of HAc added greatly exceeds that of NaOH, the solution environment has undergone significant changes. It is unclear whether this affects the surface coverage of the ligand on the nanocrystals.
5. After the addition of HAc, the particle size of nanoparticles increased from 12 nm to 18 nm, leading to a substantial reduction in specific surface area. This raises the question of whether the primary factor responsible for luminescence enhancement is H⁺ doping, or if other factors, such as reduced surface area, play a more prominent role. To better understand these effects, the author should attempt to synthesize nanoparticles with similar sizes for comparison.
6. Figure 3d reveals that the emission intensity of NMF-H-14.7 decreases after shelling, which is interesting. However, it is unclear whether this effect would have negative implications for nanoparticle applications, since coating the shell typically leads to a significant reduction in surface defect density. The author should attempt to fabricate the shell with H⁺ for comparison, with similar concentrations of H⁺ in both the core and the shell to suppress diffusion.

Reviewer #4 (Remarks to the Author):

In this paper, H⁺-doping in NaMgF₃:Yb/Er nanocrystals (NCs) were prepared with different contents of H⁺ and a three-orders-of-magnitude upconversion luminescence was detected under 980nm excitation. Authors believes that the interstitial H⁺ ions perturb the local charge density distribution and thus the crystal field around the highly symmetric Ln³⁺-substituted [MgF₆]⁴⁻ octahedron cluster, which alleviates the notorious parity-forbidden selective rule to enhance the 4f-4f radiative transition rate of the Ln³⁺ emitters. It is very interest to find a new way to enhance the upconversion luminescence by doping H⁺ ions. I recommend it to be accepted after major revisions.

1. It should be clear what is the results when H⁺ content is more than 14.7mmol? and why the particle size decreases firstly and then increases along with the increasing content of HAc?
2. In fig3, it should be clear the corresponding energy level (or the corresponding wavelength) for the lifetime measurement, in addition the UCL intensity to which energy level's emission. Have you considered the emission from 4S_{3/2}? Because the ratio of red to green changed with the different

samples, you have to describe these figures clearly.

3. Please give a clear explanation for the reason that the ratio of red to green UCL changed along with the doping content of H⁺ ions.

4. In fig4c, it is better to change the Ln to Er (emitter). I believe that H doping also affect the crystal field around Yb, but the energy level of Yb³⁺ is very simple and the absorption of Yb³⁺ increased weaker than the upconversion luminescence of Er³⁺. Therefore, the effect of H⁺ doping on the transition rate of Er³⁺ and the distance of between Yb and Er should be discussed mainly.

RESPONSE TO REVIEWERS' COMMENTS

We thank the reviewers for their critical reading and helpful suggestions. We have seriously considered their comments and revised our manuscript to address the reviewer's concerns. We have made all the requested changes that have been reflected in either manuscript or Supporting Information. All of the changes in manuscript are highlighted in **Yellow background** for clarity in the marked copy. The **blue marks** in this article are the revised parts of manuscript. The following is the point-to-point response to the comments listed in the report.

Reviewer #1

General Comment:

The manuscript is focused on demonstrating an exciting strategy to improve the emission amplification of Lanthanide ions doped nanocrystals by inserting interstitial H⁺ through the computational and experimental procedure. The hypothesis presented by the authors is that the crystal-field perturbation breaks the parity-forbidden nature of 4f-transitions of trivalent lanthanide by inserting interstitial H⁺, leading to a significant emission amplification. The work provides a relevant contribution to designing new materials with desired characteristics for several applications. The detailed discussion of the data obtained led the study to well-founded conclusions. I recommend that the article be accepted after some doubts are cleared.

Response:

We appreciate the reviewer's positive comments and thoughtful suggestions, which have been very useful in improving the quality of the manuscript. Our point-by-point responses are detailed below.

Comment #1:

How electronic states of impurities are deposited in the band gap?

Response:

Thank you for your comments. From the DFT calculations, we know that the introduction of interstitial H⁺ ions has a limited effect on the bandgap, shrinking it by only 0.173 eV. However, the corresponding density of states (DOS) showed that interstitial H⁺ contributes to the valence band, which indicates that although the effect of interstitial H⁺ on the crystal structure was limited, it still caused a change in the charge distribution within the crystal. **See the newly added Supplementary Fig. 21.**

Revised text

page 6, lines 182–185: Owing to the strong electronegativity of F⁻ ions, the interstitial H⁺ ions likely form hydrogen bonds with the nearest neighboring F⁻ ions in the [MgF₆]⁴⁻ octahedra (i.e., to form F–H···F bonds), as clearly shown in the calculated electron localization function diagram, electronic band structures and density of states (Supplementary Fig. 20 and 21).

New supplementary figures and note

Supplementary Fig. 21 | DFT-calculated electronic band structures and their corresponding projected DOS for **a**, Non-interstitial H⁺-doped NaMgF₃:Yb and **b**, Interstitial H⁺-doped NaMgF₃:Yb NCs. DFT calculated transition matrix elements for **c**, Non-interstitial H⁺-doped NaMgF₃:Yb and **d**, Interstitial H⁺-doped NaMgF₃:Yb NCs.

Supplementary Note: Effect of interstitial H⁺ on band structure and dipole transition matrix elements of NaMgF₃:Ln

DFT calculations showed that the introduction of interstitial H⁺ ions had a limited effect on the band gap, shrinking it by only 0.173 eV. However, the corresponding density of states (DOS) showed that interstitial H⁺ contributes to the valence band, which indicates that although the effect of interstitial H⁺ on the crystal structure was limited, it still caused a change in the charge distribution within the crystal (**Supplementary Fig. 21a and b**). Additionally, the effect of interstitial H⁺ ions on the transition dipole moment can be observed through DFT theoretical calculations based on first principles. Prior to the introduction of H⁺, the dipole transition matrix elements of NaMgF₃:Ln were mainly concentrated at the high-symmetry G point (**Supplementary Fig. 21c**). Therefore, the radiative transition at the 4f^N level is parity forbidden. The introduction of interstitial H⁺ produced an additional field and disturbed the original field of the crystal, resulting in the transfer of the transition dipole moment elements from the high-symmetry G point to the low-symmetry X|Y point (**Supplementary Fig. 21d**). The existence of dipole transition matrix elements with low symmetry can promote the mixing of opposite-parity configurations of lanthanide ions, thereby improving the efficiency of the Ln³⁺-emitter intra-4f optical transitions.

Comment #2:

The J-mixing effect in Ln^{3+} ions created by a distortion in local symmetry is one of the main factors that violate the Laporte rule. But the results presented by the authors don't show large distortions in the local symmetry of the lanthanide. What kind of effect is happening for this perturbation of the crystal field to occur? In my humble opinion, the entry of hydrogen may be favoring dynamic coupling. In this case, oscillating dipoles induced on the surrounding produce a field that changes the transition dipole moment of the emitter. See for example the following papers: <https://doi.org/10.1103/PhysRevLett.81.1381>; <https://doi.org/10.1016/j.jlumin.2021.118292>; [https://doi.org/10.1016/0022-3697\(95\)00020-8](https://doi.org/10.1016/0022-3697(95)00020-8); [https://doi.org/10.1016/0022-5088\(83\)90455-1](https://doi.org/10.1016/0022-5088(83)90455-1)

Response:

Thank you for your suggestion. As explained earlier, the introduction of protons (H^+) inevitably affects the charge distribution in the crystal, resulting in changes in the polarizability of the coordinated atoms around the emitter, which further affects the transition dipole moment. Therefore, it is suitable for use in dynamic coupling mechanisms. For convenience, we considered only the contributions from multipole (lanthanide: Ln)-induced dipole (ligand: L) interactions for the dynamic coupling mechanism. The Newman and Balasubramanian intensity parameters (denoted here by A_{tp}^λ) are related to the average polarizability of the ligand ($\bar{\alpha}_L$) as:

$$A_{tp}^\lambda(\text{dynamic}) = 7(-1)^p \{t(2t-1)\}^{1/2} \begin{pmatrix} 3\lambda 3 \\ 000 \end{pmatrix} \langle r^\lambda \rangle \sum_L \bar{\alpha}_L R_L^{-(t+1)} C_{-p}^t(L), \quad (1)$$

where $\lambda = 2, 4, 6, t = \lambda \pm 1, p = 0, \pm 1, \dots, \pm t$. For cases where all Ln-L pairwise interactions are cylindrically symmetric (such as $[\text{LnF}_6]^{3-}$ octahedron), A_{tp}^λ can be related to Axe's empirical $A_{tp}\Xi(t, \lambda)$ parameters on a one-to-one basis as:

$$A_{tp}^\lambda = -A_{tp}\Xi(t, \lambda) \frac{2\lambda + 1}{(2t + 1)^{1/2}}, \quad (2)$$

According to Judd–Ofelt theory, the relationship between Ω_λ and $A_{tp}\Xi(t, \lambda)$ is expressed as:

$$\Omega_\lambda = [\lambda][t]^{-1} \sum_{p,t} |A_{tp}|^2 \Xi^2(t, \lambda), \quad (3)$$

Therefore, the oscillator strength parameter Ω_λ can be considered the sum of squares of A_{tp}^λ ; according to the relationship between the electric dipole oscillator strength f_{ed} and the oscillator strength parameter Ω_λ :

$$f_{ed} = \frac{8\pi^2 mc\nu}{3h(2J+1)} \chi_{ed} \sum_{\lambda=2,4,6} \Omega_\lambda \langle 4f^N \psi_J \| U^\lambda \| 4f^N \psi'_{J'} \rangle^2, \quad (4)$$

The relationship between the electric dipole transition probability A_{ed} and electric dipole oscillator strength f_{ed} is expressed as:

$$A_{ed} = \frac{8\pi^2 e^2 \nu^2 n^2}{mc} f_{ed}, \quad (5)$$

where $m, c, \nu, h, e,$ and n are the electron mass, speed of light in vacuum, transition wavenumber, Planck constant electron charge, and crystal refractive index, respectively. The corresponding electric dipole transition probability, A_{ed} , was obtained. Therefore, the interstitial H^+ enhancing the radiation fluorescence intensity can be attributed to the fact that interstitial H^+ affects the polarizability of the ligand, and can also be described as imposing an additional field around the emitter, which promotes the

mixing of the opposite parity configurations of lanthanide ions, ultimately leading to a change in the electric dipole transition probability (A_{ed}). See newly added Refs. *J. Phys. Chem. Solids* **1995**, *56*, 1053–1062, *J. Less-Common Met.* **1983**, *93*, 113–118, *J. Lumin.* **2021**, *238*, 118292, *Phys. Rev. Lett.* **1998**, *81*, 1381–1384 and Refs. 18 (*Phys. Rev.* **1962**, *127*, 750–761), Refs. 19 (*J. Chem. Phys.* **1962**, *37*, 511–520) in the revised manuscript. See the newly added **Supplementary Methods**.

Additionally, the effect of interstitial H^+ ions on the transition dipole moment can be observed through DFT theoretical calculations based on first principles. Prior to the introduction of H^+ , the dipole transition matrix elements of $NaMgF_3:Ln$ were mainly concentrated at the high-symmetry G point. Therefore, the radiative transition at the $4f^N$ level is parity forbidden. The introduction of interstitial H^+ produced an additional field and disturbed the original field of the crystal, resulting in the transfer of the transition dipole moment elements from the high-symmetry G point to the low-symmetry X|Y point. The existence of dipole transition matrix elements with low symmetry can promote the mixing of opposite-parity configurations of lanthanide ions, thereby improving the efficiency of the Ln^{3+} -emitter intra- $4f$ optical transitions. See the newly added **Supplementary Fig. 21**.

Revised text

page 1, lines 23–26: Mechanistic studies reveal that the interstitial H^+ ions perturb the local charge density distribution, leading to the anisotropic polarization of the ligand, which affects the highly symmetric Ln^{3+} -substituted $[MgF_6]^{4-}$ octahedral cluster.

page 3 lines 76–85: The dopant H^+ ions have negligible impact on the crystal structure parameters, yet disturb the local charge density distribution and contribute to the anisotropic polarization of the ligand (F^-) of the Ln^{3+} emitter^{1,2}. This introduces an additional field in the local structure around the Ln^{3+} emitter and facilitates the mixing of odd-parity states into the $4f$ wavefunction^{2,3,4}. According to the Judd–Ofelt principle^{2,5,6,7,8}, the mixing of opposite-parity Ln^{3+} configurations enhances the oscillator strength of the electric dipole transition (f_{ed})². Since the electric dipole radiative transition probability (A_{ed}) is positively correlated with f_{ed} ($A_{ed} \propto f_{ed}$), such that an elevation of f_{ed} is accompanied by an increase in the A_{ed} of intra- $4f$ optical transitions, which ultimately transforms the dim luminescence emission caused by high crystal symmetry to bright luminescence emission (Fig. 1a and Supplementary Methods).

page 6 lines 187–195: The $F-H\cdots F$ bonds would exert a slight effect on the local coordination environment around the Ln^{3+} emitter in the substituted $[MgF_6]^{4-}$ octahedra—that is, crystal-field perturbation—by nonequivalently changing the bond lengths, angles (Fig. 4c), and differential charge density distributions (Fig. 4d), which implied that the ligand (F^-) of Ln^{3+} was anisotropically polarised at the presence of interstitial H^+ . In this case, oscillating dipoles induced on the surrounding led to additional ligand-field interactions that promote mixing of opposite-parity states of Ln^{3+} emitter, thus partially breaking the parity-forbidden nature of the intra-configurational $4f$ transitions. The change of dipole transition matrix element can also prove this point (Supplementary Fig. 21).

page 6, lines 190–195: This implies that the ligands (F^-) of the Ln^{3+} ions are anisotropically polarized in the presence of interstitial H^+ ions. The oscillating dipoles induced in the local structure lead to additional ligand–field interactions that promote the mixing of opposite-parity states of the Ln^{3+} emitter, thus partially breaking the parity-forbidden nature of the intra-configurational $4f$ transitions.^{1,2} The change in the transition dipole moment further proves this point (Supplementary Fig. 21).

page 8–9 lines 271–272: ...interstitial H^+ -doping exerts a crystal-field perturbation effect by inducing

the anisotropic polarization of the ligand (F^-), which partially mitigates the fundamental constraint of parity-forbidden $4f-4f$...

New added references

23. MALTA, O. L. The theory of vibronic transitions in rare earth compounds. *J. Phys. Chem. Solids* **56**, 1053–1062 (1995).
37. Souza, A. S., Cortes, G. K., Lima, H. & Santos, M. A. C. d. The local-field correction factor beyond the Onsager–Böttcher approach: Mixing of states from the interaction with atoms in the surrounding medium. *Journal of Luminescence* **238**, 118292 (2021).
38. Reid, M. F. & Richardson, F. S. Rationalization of the f-f intensity parameters for transitions between crystal field levels of lanthanide ions. *J. Less-Common Met.* **93**, 113-118 (1983).
39. Vries, P. d. & Legendijk, A. Resonant scattering and spontaneous emission in dielectrics: microscopic derivation of local-field effects. *Phys. Rev. Lett.* **81**, 1381–1384 (1998).

Comment #3:

Did the authors verify the defect clustering with the lanthanide entering into the Na site by DFT calculations? What is the defect formation energy?

Response:

Thank you for your comments. We verified that lanthanides entered the Na sites using previous DFT calculations. The calculation results showed that the formation energy of lanthanide ions replacing Na^+ ions was considerably larger than that of lanthanide ions replacing Mg^{2+} ions owing to the similar valences and ionic radii of Mg^{2+} and Ln^{3+} . See also Refs. 29 (*Mater. Adv.*, **2021**, 2, 1378–1389).

Regarding the comment “What is the defect formation energy?”. In our DFT calculations, the formation energies required for Ln^{3+} to replace Na^+ and generate two Na vacancies was -0.514 eV/atom. The formation energy required for Ln^{3+} to replace Na^+ and generate one Mg vacancy was -0.507 eV/atom, respectively. The formation energy required for these two substitution methods was much higher than that required for Ln^{3+} to replace Mg^{2+} and generate one Na vacancy, thereby indicating that Ln^{3+} tends to enter the Mg sites in $NaMgF_3$ crystals. See the newly added **Supplementary Fig. 3**.

New supplementary figures

Supplementary Fig. 3 | Formation energy (ΔE_{form}) per atom for **a**, $NaMgF_3:H$, **b**, $NaMgF_3:Yb$, and **c**,

NaMgF₃:Yb/H crystals, as determined by density functional theory (DFT) calculations. H⁺_{Na} represents H⁺ in a Na⁺ site, Yb³⁺_{Na} represents Yb³⁺ in a Na⁺ site, Yb³⁺_{Mg} represents Yb³⁺ in a Mg²⁺ site, V_{Na} is a Na⁺ vacancy, and H_i is interstitial H⁺. The results show that, for NaMgF₃ and NaMgF₃:Yb, interstitial H⁺ doping (NaMgF₃:H_i in **a** and NaMgF₃:Yb³⁺_{Mg}-V_{Na}-H_i in **c**) has a lower formation energy than other doping behaviors. In addition, Yb will preferentially occupy the Mg lattice sites (NaMgF₃:Yb³⁺_{Mg}-V_{Na} in **b**) rather than the Na ones (NaMgF₃:Yb³⁺_{Na}-V_{Mg} in **b**).

Thank you again for your careful review and constructive comments and advice for improving our manuscript. We would be glad to respond to any further questions and comments that you may have, or make further changes, if required.

Reviewer #2

General Comment:

This manuscript reports the synthesis of trivalent lanthanide-based nanocrystals (NCs) with the formula $\text{NaMgF}_3:\text{Ln}^{3+}$ or NMF-H-X , where X is the acid concentration (mmol), $\text{Ln}^{3+} = \text{Yb}^{3+}$ and Er^{3+} , and NaMgF_3 crystals are used as a model host matrix. Due to parity-forbidden $4f-4f$ transitions, these NCs exhibit weak upconversion (UC) and downconversion (DC) luminescence. The authors found that the interstitial H^+ -doping with acetic acid, formic acid, hydrochloric acid, and benzenesulfonic acid (BL70) can drastically improve the UC and DC luminescence intensities of these NCs. After using BL70 as a proton source, NMF-H-0.5 displayed an increase in UC luminescence intensity of 1890.6-fold. The authors hypothesized that crystal-field perturbation, which aids in breaking the parity-forbidden nature of $4f$ -transitions in these NCs, causes this sharp increase in UC luminescence intensities after H^+ -doping. Finally, H^+ -doped $\text{NaMgF}_3:\text{Yb/Er}$ NCs with NIR-II emission was used to perform high-resolution angiography on nude mice.

Overall, the work seems good, but several unanswered questions must be explained. I recommend that the manuscript be accepted by "Nature Communications" after major revision. The detailed comments are shown below.

Response:

We thank the reviewer for their positive comments and thoughtful suggestions. Point-by-point responses to all comments are provided below.

Comment #1:

Do the conjugate bases, such as chloride ion (Cl^-), formate ion (HCOO^-), or benzenesulfonate (PhSO_3^-), affect the size of the NCs? Can conjugate base PhSO_3^- affect the parity-forbidden $4f$ transitions?

Response:

Thank you for your comment. We synthesized $\text{NaMgF}_3:\text{Yb/Er}$ NCs using NaCl , HCOONa and PhSO_3Na as Na^+ sources and found that the synthesized NCs had non-uniform morphology and poor dispersion. However, the addition of the acids corresponding to these conjugated bases (HCl , HCOOH , PhSO_3H) can not only synthesize uniform $\text{NaMgF}_3:\text{Yb/Er}$ NCs, but also improve the luminescence of $\text{NaMgF}_3:\text{Yb/Er}$ NCs like HAc . See the newly added **Supplementary Fig. 28–32**.

Regarding the comment “*Can conjugate base PhSO_3^- affect the parity-forbidden $4f$ transitions*”, we did not find any relevant literature reports on whether PhSO_3^- affects parity-forbidden $4f$ transitions. In our study, we found that the conjugate base PhSO_3^- does affect the crystal morphology, which in turn can affect the luminescence intensity of the NCs. See the additional **Figure S1** attached below.

Additional Figure S1 | Comparison of typical UCL spectra of NMF-H-0 (with NaOH), NMF-H-0 (with PhSO₃Na), and NMF-H-0.5 (with NaOH and PhSO₃H (BL70)) under 980 nm diode laser excitation with a power density of 50 W/cm².

Revised text

page 7, lines 227-231: In another set of experiments, we prepared interstitially H⁺-doped NaMgF₃:Yb/Er NCs with different H⁺ sources (formic acid, hydrochloric acid, propionic acid, and benzenesulfonic acid (PhSO₃H; BL70); see Methods). Er³⁺ UCL enhancements were achieved in all cases (Supplementary Figs. 28–31), with the BL70 precursor providing the largest enhancement factor of 1891 (Fig. 5b and Supplementary Fig. 31).

page 7, lines 233-235: Interestingly, the conjugated bases corresponding to these acids deteriorate the crystal morphology, whereas H⁺ can stabilize the crystal morphology (Supplementary Fig. 32).

New and revised supplementary figures

Supplementary Fig. 28 | Effect of formic acid (HCOOH) as a H^+ source on the $NaMgF_3:Yb/Er$ NC size, morphology, and UCL intensity. **a**, TEM images of NMF-H- X (with HCOOH) NCs with $X = 1.59, 2.65,$ and 3.98 mmol nominal HCOOH (Na^+ source: NaOH). The NC size was calculated from the sizes of 100 NCs in typical TEM images, and is given as the mean \pm standard deviation at the bottom of each panel in **a**. Scale bars in **a**: 50 nm. **b**, UCL spectra and **c**, corresponding enhancement factors of NMF-H- X (with HCOOH) NCs (nominal amounts of 0–3.98 mmol HCOOH) under 980 nm diode laser excitation with a power density of 50 W/cm². The maximum UCL enhancement factor was 186.2 at 2.65 mmol HCOOH.

Supplementary Fig. 29 | Effect of HCl as a H^+ source on the $NaMgF_3:Yb/Er$ NC size, morphology, and UCL intensity. **a**, TEM images of NMF-H- X (with HCl) NCs with $X = 0.18, 0.37, 0.61,$ and 1.22 mmol nominal HCl (Na^+ source: NaOH). The NC size was calculated from the sizes of 100 NCs in typical TEM images, and was given as the mean \pm standard deviation at the bottom of each panel in **a**. Scale bars in **a**: 50 nm. **b**, UCL spectra and **c**, corresponding green ($^2H_{11/2}, ^4S_{3/2} \rightarrow ^4I_{15/2}$ of Er) and red ($^4F_{9/2} \rightarrow ^4I_{15/2}$ of Er) UCL enhancement factors of NMF-H- X (with HCl) NCs (nominal amounts of 0–1.22 mmol HCl). The inset in **c** showed the enhancement factor of the integrated UCL intensity from 400 to 750 nm. The maximum UCL enhancement factor was 141.5 at 1.22 mmol HCl.

Supplementary Fig. 30 | Effect of propionic acid (PA) as a H^+ source on the $NaMgF_3:Yb/Er$ NC size, morphology, and UCL intensity. **a**, TEM images of NMF-H- X (with PA) NCs with $X = 1.6, 3.1,$ and 6 mmol nominal PA (Na^+ source: $NaOH$). The NC size was calculated from the sizes of 100 NCs in typical TEM images, and is given as the mean \pm standard deviation at the bottom of each panel in **a**. Scale bars in **a**: 50 nm. **b**, UCL spectra and **c**, corresponding enhancement factors of NMF-H- X (with PA) NCs (nominal amounts of 0 – 6 mmol PA) under 980 nm diode laser excitation with a power density of 50 W/cm^2 . The maximum UCL enhancement factor was 353.4 at 6 mmol PA.

Supplementary Fig. 31 | Effect of benzenesulfonic acid (BL70) as a H⁺ source on the NaMgF₃:Yb/Er NC size, morphology, and UCL intensity. **a**, TEM images of NMF-H-*X* (with BL70) NCs with *X* = 0.025, 0.05, 0.1, 0.2, 0.5, and 0.8 mmol BL70 (Na⁺ source: NaOH). The NC size was calculated from the sizes of 100 NCs in typical TEM images, and is given as the mean ± standard deviation at the bottom of each panel in **a**. Scale bars in **a**: 50 nm. **b**, UCL spectra and **c**, corresponding enhancement factors of NMF-H-*X* (with BL70) NCs (nominal amounts of 0–0.8 mmol BL70) under 980 nm laser excitation with a power density of 50 W/cm². The maximum UCL enhancement factor was 1890.6 at 0.5 mmol BL70.

Supplementary Fig. 32 | TEM images of NaMgF₃:Yb/Er (NMF-H-0) NCs synthesized with different Na⁺ sources: **a**, NaCl, **b**, HCOONa, and **c**, PhSO₃Na. The morphologies of these NCs were all different, indicating that the use of different conjugate bases (Cl⁻, HCOO⁻, and PhSO₃⁻) results in NCs with different morphologies. Among them, Cl⁻ and PhSO₃⁻ have a negative effect on the NC morphology, with the NCs exhibiting undefined shapes and agglomeration.

Comment #2:

What makes the shapes of octahedra and dodecahedra appropriate for interstitial H⁺-doping?

Response:

We apologize for the misunderstanding. We have re-read the manuscript carefully and believe this confusion stems from lines 89–91: “NaMgF₃ has rich lattice voids created by neighboring [MgF₆]⁴⁻ octahedra and [NaF₈]⁷⁻ dodecahedra, which are believed to facilitate interstitial H⁺-doping.” In fact, what we want to express in this sentence is that the abundant lattice voids of NaMgF₃ facilitate interstitial H⁺ doping. The lattice voids happen to be within regular [MgF₆]⁴⁻ octahedra and malformed [NaF₈]⁷⁻ dodecahedra. See also Refs. 9 (*Norway. J. Mineral. Geochem.* **2005**, 182, 23–29) in Supplementary Information; however, it is the lattice voids formed by the octahedral and dodecahedral shapes, as opposed to the shapes themselves, that promote interstitial H⁺ doping. The DFT calculations in the original manuscript also show that H⁺ readily entered the NaMgF₃ crystal by interstitial doping. The ortho-hexahedral structure ([CaF₈]⁶⁻) of CaF₂ crystals also has lattice voids and is therefore also capable of achieving a substantial increase in UCL intensity by interstitial H⁺ doping. To avoid the same confusion for other readers, we have revised original statement to “NaMgF₃ has rich lattice voids that are believed to facilitate interstitial H⁺-doping.” The crystal structure explanation “lattice voids created by neighboring [MgF₆]⁴⁻ octahedra and [NaF₈]⁷⁻ dodecahedra” has been transferred to the new Supplementary Note (page 6, lines 88–89). In addition, the octahedral and dodecahedral structures and included lattice voids are shown in the newly added Supplementary Fig. 2.

Revised text

page 3, lines 89–90: NaMgF₃ has rich lattice voids that are believed to facilitate interstitial H⁺-doping (Fig. 1b and Supplementary Table 1 and Fig. 2).

Revised supplementary text

page 6, lines 87–88: The neighboring [MgF₆]⁴⁻ octahedra and [NaF₈]⁷⁻ dodecahedra create an abundance

of lattice voids¹⁵.

New supplementary figure

Supplementary Fig. 2 | Schematic diagram of the sublattice structure of NaMgF₃. Na⁺ ions fill dodecahedral cavities composed of eight [MgF₆]⁴⁻ *ortho*-octahedra. However, Na⁺ is smaller than the dodecahedral cavity and coordinates with the neighboring eight F⁻ anions to form distorted [NaF₈]⁷⁻ dodecahedra. The four F⁻ anions at the distal sites do not contribute to the coordination of Na⁺, resulting in the formation of a cavity structure (lattice void) by the four distal F⁻ anions. This cavity structure facilitates interstitial doping.

Comment #3:

In the case of BL70, why is the UC luminescence intensity of NMF-H-0.5 found to be higher than that of NMF-H-0.8?

Response:

Thank you for your comment. The UCL intensity reaches a maximum as the amount of doped interstitial H⁺ increases (for BL70, around NMF-H-0.5). This is because, on one hand, H⁺ ions have a positive effect on emission by breaking the parity-forbidden 4*f* transitions, while on the other hand, interstitial H⁺ ions are essentially interstitial atomic defects, which means that excessive H⁺ doping inevitably causes lattice distortion and reduces luminescence intensity. These two factors jointly affect the luminescence intensity. In fact, this effect is observed for HAc as well as BL70 (see newly added **Supplementary Fig. 9** and **Supplementary Note** below). The threshold for lattice distortion was reached with a lower nominal addition of BL70 (0.5 mmol) compared to that for HAc (≤14.7 mmol), because BL70 is a much stronger organic acid and releases more H⁺ during the reaction for interstitial H⁺ doping. Therefore, when the nominal amount of BL70 exceeded 0.5 mmol, the number of interstitial H⁺ caused lattice distortion that led to weakening of the fluorescence intensity.

New supplementary figure

Supplementary Fig. 9 | **a**, Typical UCL spectra, **b**, TEM images, and **c**, corresponding high-resolution TEM (HRTEM) images of as-synthesized NaMgF₃:Yb/Er NCs with excess HAc added during the synthetic procedure (14.7–25 mmol) (Na⁺ source: NaOH). The inset in **a** showed the UCL intensity as a function of HAc content. The TEM images in **b** showed that excessive interstitial H⁺ doping alters the NaMgF₃:Yb/Er NC morphology. The Fourier-filtered HRTEM images in the insets in **c** reveal that excessive interstitial H⁺ doping led to serious lattice distortion, which may be the main reason for the decrease of crystal fluorescence intensity.

New supplementary note: Lattice distortion at high interstitial H⁺-doping content

The luminescence intensity of NaMgF₃:Yb/Er NCs reached a maximum when the nominal amount of HAc was 14.7 mmol, and further increased in the HAc addition led to a decrease in the luminescence intensity and change of crystal morphology (**Supplementary Fig. 9**). Because interstitial H⁺ ions were essentially interstitial atomic defects, large amounts of H⁺ doping led to lattice distortion (e.g., changes in the lattice fringes, as shown in **Supplementary Fig. 9c**). Severe lattice distortion can cause fluorescence quenching and a reduction in luminescence intensity. Therefore, only a small amount of HAc (≤14.7 mmol) was added to the reaction solvent to illustrate the crystal field perturbation effect.

Comment #4:

Why does the addition of 1.6 mmol of acetic acid decrease the crystal size from 12.6 Å to 6.7 Å, but the addition of 3.1 mmol of acetic acid increases the crystal size to 12.3 Å?

Response:

Thank you for this question. This may be related to variations in the NaMgF₃:Ln³⁺ formation energy as the amount of interstitial H⁺ changes. DFT calculations showed that the formation energy increases

slightly with small amounts of interstitial H⁺ doping, but further increased in the interstitial H⁺ content result in a significant decrease in the required formation energy. Higher formation energies mean that the substances are more difficult to form; therefore, after the initial increase in formation energy for small amounts of interstitial H⁺ doping, NMF-H-1.6 NCs (which have a higher formation energy) had a smaller particle size than NCs with higher amounts of interstitial H⁺ (which had a lower formation energy), assuming the reaction time stays constant (as was the case here). This discussion has been added to the manuscript as a new Supplementary Note (“Crystal size of NaMgF₃ NCs with different HAc precursor additions”, page 6, lines 136–143).

New supplementary note: Crystal size of NaMgF₃ NCs with different HAc precursor additions

The crystal size of the interstitially H⁺-doped NaMgF₃ NCs first decreased and then increased with increasing interstitial H⁺ content (Supplementary Fig. 6). This may be related to the variation of the formation energy (ΔE_{form}) of NaMgF₃:Ln³⁺ with the amount of interstitial H⁺. DFT calculations showed that ΔE_{form} increases slightly with small amounts of interstitial H⁺ doping, but further increases in the amount of interstitial H⁺ result in a significant decrease in ΔE_{form} (Fig. 1c). Higher ΔE_{form} values mean that the corresponding substances are more difficult to form; therefore, in the same reaction time, NCs with a higher ΔE_{form} (e.g., NMF-H-1.6) form with a smaller particle size than those with a lower ΔE_{form} (e.g., NCs with increased amounts of interstitial H⁺).

Comment #5:

Why does the UC emission intensity of the $^4S_{3/2} \rightarrow ^4I_{15/2}$ transition have no effect when using acetic acid as a proton source but changes in the presence of HCl?

Response:

We apologize for the unclear spectral data. Both HAc and HCl H⁺ sources can effectively enhance the UCL intensity of the $^4S_{3/2} \rightarrow ^4I_{15/2}$ transition. Specifically, the green UCL intensity of NaMgF₃:Yb/Er NCs increased by up to 258 times when using HAc and 132.2 times when using HCl. We have added the data for HAc and HCl as newly added Supplementary Figs. 10a and 29b and c to ensure this is clearly presented.

Revised text

page 4, lines 131–134: Indeed, the overall UCL intensity of the NMF-H-14.7 NCs was up to 675 times higher than that of NMF-H-0 (Fig. 3b and Supplementary Fig. 9), with green and red UCL intensity enhancement factors of 258 and 706.3, respectively (Supplementary Fig. 10).

New supplementary figures

Supplementary Fig. 10 | a, Green and **b**, Red UCL spectra (450–600 nm) of NaMgF₃:Yb/Er NCs synthesized with HAc (nominal amounts of 0–14.7 mmol) (Na⁺ source: NaOH) under 980 nm diode laser excitation with a power density of 50 W/cm². The insets in **a** and **b** showed the corresponding enhancement factors of the green and red UCL intensities for NMF-H-X (with HAc). The interstitial H⁺ doping strategy can effectively enhanced the green and red UCL intensities simultaneously, with enhancement factors of up to 258 and 706.3 for green (²H_{11/2}, ⁴S_{3/2}→⁴I_{15/2}) and red (⁴F_{9/2}→⁴I_{15/2}) emissions, respectively.

Supplementary Fig. 29 | Effect of HCl as a H^+ source on the $NaMgF_3:Yb/Er$ NC size, morphology, and UCL intensity. (Na^+ source: NaOH) **a**, TEM images of NMF-H- X (with HCl) NCs with $X = 0.18, 0.37, 0.61,$ and 1.22 mmol nominal HCl. The NC size was calculated from the sizes of 100 NCs in typical TEM images, and was given as the mean \pm standard deviation at the bottom of each panel in **a**. Scale bars in **a**: 50 nm. **b**, UCL spectra and **c**, corresponding green ($^2H_{11/2}, ^4S_{3/2} \rightarrow ^4I_{15/2}$ of Er) and red ($^4F_{9/2} \rightarrow ^4I_{15/2}$ of Er) UCL enhancement factors of NMF-H- X (with HCl) NCs (nominal amounts of 0–1.22 mmol HCl). The inset in **c** showed the enhancement factor of the integrated UCL intensity from 400 to 750 nm. The maximum UCL enhancement factor was 141.5 at 1.22 mmol HCl.

Comment #6:

Comparing the UC luminescence intensities of NMF-H-0, NMF-H-14.7, NMF-H-0 (with NaOAc), and NMF-H-0 (with NaOH) is recommended.

Response:

Thank you for this suggestion. We compared the UCL intensities of NMF-H-0 (with NaAc), NMF-H-0 (with NaOH), and NMF-H-14.7 (with NaOH and HAc) according to your suggestion (see newly added **Supplementary Fig. 16**). Because the fluorescence intensity of the NMF-H-14.7 NCs was much stronger than that of the NMF-H-0 (with NaAc or NaOH) NCs, under identical measurement conditions (as in Supplementary Fig. 14), the emission spectra for NMF-H-0 (with NaOH or NaAc) was of poor quality due to the weak UCL fluorescence signals. Nevertheless, their comparing the UCL intensities showed that the Ac^- ions do not enhance the fluorescence intensity of $NaMgF_3:Yb/Er$ upconversion, and the interstitial H^+ ions are key to amplifying the upconversion fluorescence signal.

New supplementary figure

Supplementary Fig. 16 | Comparison of the typical UCL spectra of NMF-H-14.7 (with NaOH and HAc),

NMF-H-0 (with NaAc), and NMF-H-0 (with NaOH) NCs under 980 nm diode laser excitation with a power density of 50 W/cm². Owing to the weak luminescence intensity of the NMF-H-0 samples without interstitial H⁺ doping, the inset showed an enlarged view of the NMF-H-0 (with NaAc or NaOH) spectra below the dashed line. The UCL intensity of NMF-H-0 (with NaAc) was slightly stronger than that of NMF-H-0 (with NaOH), because NaOH inevitably introduced trace amounts of OH⁻ anions to the NCs, which quench the luminescence of lanthanide ions to some extent.

Comment #7:

Why is the UCL intensity of NMF-H-0 (with NaOAc) higher than NMF-H-0 (with NaOH)?

Response:

Thank you for your question. The synthesis of NaMgF₃:Yb/Er NCs with NaOH inevitably introduce a small amount of OH⁻ anions which will somewhat quench the luminescence of the lanthanide ions. Therefore, the UCL intensity of NMF-H-0 (with NaAc) was slightly stronger than that of NMF-H-0 (with NaOH). This information was added to the caption of **Supplementary Fig. 16** (page 24, lines 274–276).

Comment #8:

What role does NaOH play in synthesizing NMF-H-0 NC? Can NaOAc be used instead?

Response:

Thank you for this interesting question. The synthesis of NaMgF₃ by NaOH can ensure a perfect crystal morphology and good dispersion in cyclohexane for subsequent experiments. In our experimental design, NaOH and ammonium fluoride (NH₄F) were directly blended in a methanol solution with a molar ratio of 1:1. NaOH and NH₄F reacted to produce NH₃·H₂O and NaF in solution, but NH₃·H₂O volatilized during the methanol evaporation step (evaporation temperature: 70–80 °C), which further promoted the formation of NaF. NaF acted as crystal nucleus for the formation of NaMgF₃ and therefore promoted the formation of NaMgF₃ crystals.

Regarding the comment “*Can NaOAc be used instead*”. The interstitial H⁺-doping strategy was also realized when using sodium acetate (NaAc) instead of NaOH (see newly added **Supplementary Fig. 17a**). However, when using NaAc, the crystal morphology and sample dispersion cannot be effectively guaranteed (see newly added **Supplementary Fig. 17b**), which was not conducive to the relevant research work. Therefore, we mainly used NaOH to synthesize NaMgF₃ NCs in the present work after comprehensive consideration.

Revised text

page 5, lines 163–168: Notably, introducing H⁺ ions to the NMF-H-X NCs synthesized with NaAc resulted in a gradual increase in UCL intensity (**Supplementary Fig. 17a**), although the synthesized samples had a worse crystal morphology than those synthesized with NaOH (**Supplementary Fig. 17b**).

New supplementary figure

Supplementary Fig. 17 | **a**, Typical UCL spectra of $\text{NaMgF}_3\text{:Yb/Er}$ (with NaAc) NCs as a function of the nominal amount of HAc ($X = 0\text{--}14.7$ mmol) added during synthesis. The inset shows the corresponding UCL enhancement factors for NMF-H- X (with NaAc and HAc). The increasing UCL intensity with HAc content indicates that changing the Na^+ source does not affect the implementation of the interstitial H^+ doping strategy. **b**, TEM images of NMF-H-0 NCs synthesized with NaAc (left) and NaOH (right). Although the interstitial H^+ doping strategy can also be achieved by using NaAc as the Na^+ source, good crystal morphology cannot be obtained. Scale bars in **b**: 50 nm.

Comment #9:

The quantum yield of all samples should be measured and reported in the manuscript.

Response:

Thank you for this suggestion. We measured the UC and NIR quantum yield (QY) of the NMF-H- X NCs ($X = 0, 1.6, 2.6, 3.1, 7.3$, and 14.7 mmol HAc). The UCQYs of NMF-H-14.7, NMF-H-7.3, NMF-H-3.1, and NMF-H-2.6 were 0.18%, 0.165%, 0.036%, and 0.029%, respectively; and the NIR-QYs were 20.87%, 15.87%, 8.99%, and 8.23%, respectively (see newly added **Supplementary Table 5**). The UCQY and NIR-QY data for NMF-H-0 and NMF-H-1.6 were recorded as $<0.01\%$, as their fluorescence signals did not reach the lower detection limit of the instrument.

Revised text

page 4, lines 128–131: Notably, despite the extremely low Yb/Er content (≤ 2.1 mol%, Supplementary Table 4), the overall upconversion quantum yield (UCQY) of the interstitially H^+ -doped $\text{NaMgF}_3\text{:Yb/Er}$ NCs increased from $<0.01\%$ to 0.18% in the presence of interstitial H^+ (Supplementary Table 5), resulting in luminescence that was visible to the naked eye.

New supplementary table

Supplementary Table 5 | Upconversion- and NIR-quantum yield (UCQY and NIR-QY) of NMF-H- X ($X = 0, 1.6, 2.6, 3.1, 7.3$, and 14.7 mmol HAc) NCs.

Parameter	Sample					
	NMF-H-0 ^a	NMF-H-1.6 ^a	NMF-H-2.6	NMF-H-3.1	NMF-H-7.3	NMF-H-14.7
UCQY (%)	<0.01	<0.01	0.029	0.036	0.165	0.180
NIR-QY (%)	<0.01	<0.01	8.23	8.99	15.87	20.87

^a The UCQY and NIR-QY data for NMF-H-0 and NMF-H-1.6 were below the detection limit of the instrument.

Comment #10:

Can temperature affect the ${}^4F_{9/2} \rightarrow {}^4I_{15/2}$ transition?

Response:

Yes, temperature can affect the ${}^4F_{9/2} \rightarrow {}^4I_{15/2}$ transition. According to your suggestion, we measured the UCL intensity of the ${}^4F_{9/2} \rightarrow {}^4I_{15/2}$ transition for NMF-H-0, NMF-H-3.1, and NMF-H-14.7 NCs at temperatures between -190 and 300 °C (see newly added **Supplementary Fig. 26**). The general trend was that the emission intensity decreased with temperature; however, interestingly, NMF-H-0 and NMF-H-3.1 both exhibited thermal enhancement of the UCL intensity between 50 and 150 °C. This is because the Ln^{3+} ($\text{Yb}^{3+}/\text{Er}^{3+}$) dopant occupies the Mg^{2+} lattice sites in the NaMgF_3 crystals, and therefore have highly symmetrical crystal field environments (S_6 symmetry). The crystal cell undergoes thermal expansion upon increasing the temperature, which reduces the positional symmetry of Ln^{3+} and therefore partially alleviates the parity-forbidden selection rule that is strictly observed in high symmetry environments. Alleviation of the parity-forbidden selection rule thus facilitates Ln^{3+} -emitter intra- $4f$ optical transitions and enhances the photoluminescence emission intensity (*Chem. Mater.* **2021**, *33*, 158–163). Both NMF-H-3.1 and NMF-H-0 showed similar thermal UCL enhancement, indicating that the sublattice structure of Ln^{3+} within them is similar. This further supports the notion that interstitial H^+ -doping only causes crystal field perturbation in the crystal field environment around Ln^{3+} , and does not change the corresponding crystal structure. The more pronounced thermal UCL enhancement for NMF-H-3.1 compared to that for NMF-H-0 is related to the extremely weak UCL intensity of the NMF-H-0 NCs, which makes the detection of emission signals difficult. Interestingly, the NMF-H-14.7 NCs do not exhibit this UCL thermal enhancement phenomenon (**Supplementary Fig. 26c**), indicating that extensive interstitial H^+ -doping may reduce the sublattice symmetry. This discussion has been added to the manuscript as a new **Supplementary Note** (“**Temperature-dependence of UCL of interstitially H^+ -doped $\text{NaMgF}_3:\text{Yb}/\text{Er}$ NCs**”, pages 39–40, lines 428–447).

Revised text

page 7, lines 217–220: The similarity in the temperature-dependence of the UCL spectra of the NMF-H-0 and NMF-H-3.1 NCs (**Supplementary Fig. 26**) indicates that Er^{3+} remains in the center of the highly symmetric $[\text{MgF}_6]^{4-}$ octahedra, which further confirms the crystal-field perturbation effect exerted by interstitial H^+ -doping.

New added supplementary reference

16. Ren, B. et al. Synthesis of core-shell ScF₃ nanoparticles for thermal enhancement of upconversion. *Chem. Mater.* **33**, 158–163 (2021).

New supplementary figure

Supplementary Fig. 26 | Temperature-dependence (−190 to 300 °C) of red UCL intensity (${}^4F_{9/2} \rightarrow {}^4I_{15/2}$ transition) of **a**, NMF-H-0, **b**, NMF-H-3.1, and **c**, NMF-H-14.7 NCs (H^+ source: HAc; Na^+ source: NaOH) under 980 nm laser excitation. The insets show an enlargement of the region between 50 and 150 °C.

Supplementary Note: Temperature-dependence of UCL of interstitially H⁺-doped NaMgF₃:Yb/Er NCs

UCL measurements at different temperatures (**Supplementary Fig. 26**) show that the red UCL intensity generally decreases with increasing temperature. However, both NMF-H-0 and NMF-H-3.1 exhibit slight thermal enhancement of the UCL intensity between 50 and 150 °C (**Supplementary Figs. 26a and 26b**). This is because the Ln³⁺ (Yb³⁺/Er³⁺) dopants occupy the Mg²⁺ lattice sites in NaMgF₃, and therefore have a highly symmetrical crystal field environment (*S*₆ symmetry). The crystal cell undergoes thermal expansion upon increasing the temperature, which reduces the positional symmetry of Ln³⁺ and therefore partially alleviates the parity-forbidden selection rule that is strictly observed in high symmetry environments. Alleviation of the parity-forbidden selection rule facilitates Ln³⁺-emitter intra-4*f* optical transitions and enhances the photoluminescence emission intensity¹⁰. Both NMF-H-3.1 and NMF-H-0 showed similar thermal enhancement of UCL, indicating that the sublattice structure of Ln³⁺ within them is similar. This further supports the notion that interstitial H⁺-doping only causes crystal field perturbation in the crystal field environment around Ln³⁺, and does not change the corresponding crystal structure. The more pronounced thermal enhancement of UCL for NMF-H-3.1 compared to that for NMF-H-0 is

related to the extremely weak UCL intensity of the NMF-H-0 NCs, which makes the detection of emission signals difficult. Interestingly, the NMF-H-14.7 NCs do not exhibit this UCL thermal enhancement phenomenon (**Supplementary Fig. 26c**), indicating that extensive interstitial H⁺-doping has some effect on the sublattice symmetry.

Comment #11:

Why did the authors choose acetic acid, formic acid, hydrochloric acid, and benzenesulfonic acid (BL70) as proton sources?

Response:

Thank you for this question. There are three main reasons for why HAc was chosen as the proton source: 1. The Mg²⁺, Yb³⁺, and Er³⁺ precursors were all acetate solutions; therefore, the use of HAc as a proton source reduces the influence of other anions; 2. HAc is a weak acid; therefore, the crystal field of NaMgF₃:Yb/Er can be fine-tuned by slowly changing the amount of interstitial H⁺, so as to understand the effect of crystal-field perturbation on the emission intensity of NaMgF₃:Yb/Er; and 3. HAc is a relatively safe proton source and there is no risk of adverse effects to the researchers in the event of gas spillage during the reaction.

The other acids mentioned in the original manuscript, such as formic acid, hydrochloric acid, and phenolic acid, were randomly selected. The purpose of these experiments was to prove that different kinds of acids (which can effectively provide H⁺) are all capable of achieving our proposed H⁺-doping strategy. To further illustrate the effect of the H⁺ source, we randomly selected another H⁺ source (propionic acid (PA)), and repeated the experiments with it. PA also greatly improved the emission intensity of NaMgF₃:Yb/Er (PA: 353.4-fold; see newly added **Supplementary Figs. 30**).

Revised text

page 7, lines 227–231: In another set of experiments, we prepared interstitially H⁺-doped NaMgF₃:Yb/Er NCs with different H⁺ sources (formic acid, hydrochloric acid, propionic acid, and benzenesulfonic acid (PhSO₃H; BL70); see Methods). Er³⁺ UCL enhancements were achieved in all cases (Supplementary Figs. 28–31), with the BL70 precursor providing the largest enhancement factor of 1891 (Fig. 5b and Supplementary Fig. 31).

New supplementary figure

Supplementary Fig. 30 | Effect of propionic acid (PA) as a H^+ source on the $NaMgF_3:Yb/Er$ NC size, morphology, and UCL intensity. **a**, TEM images of NMF-H- X (with PA) NCs with $X = 1.6, 3.1,$ and 6 mmol nominal PA. The NC size was calculated from the sizes of 100 NCs in typical TEM images, and is given as the mean \pm standard deviation at the bottom of each panel in **a**. Scale bars in **a**: 50 nm. **b**, UCL spectra and **c**, corresponding enhancement factors of NMF-H- X (with PA) NCs (nominal amounts of 0–6 mmol PA) under 980 nm diode laser excitation with a power density of 50 W/cm^2 . The maximum UCL enhancement factor was 353.4 at 6 mmol PA.

Comment #12:

The authors are encouraged to change the word “proof” to “report” in the “Abstract” section.

Response:

Thank you very much for your suggestion. We carefully examined the “Abstract” section of the original manuscript and did not find the word “proof”, but we changed “prove” to “report”.

Revised text

page 1, line 21–23: Herein, we report crystal-field perturbation through interstitial H^+ -doping in orthorhombic-phase $NaMgF_3:Ln^{3+}$ NCs and achieve a three-orders-of-magnitude emission amplification without distinct lattice distortion.

Comment #13:

The phrase “see Methods online” is used frequently by the authors. In this regard, kindly provide the appropriate references.

Response:

We apologize for the misunderstanding. The phrase “*see Methods online*” in the manuscript refers to the “*Methods*” section. To reduce misunderstandings, we have revised “*see Methods online*” to “*Methods.*”

Thank you again for your careful review and constructive comments and advice for improving our manuscript. We would be glad to respond to any further questions and comments that you may have, or make further changes, if required.

Reviewer #3

General Comment:

This paper entitled “Proof of Crystal-Field-Perturbation-Enhanced Luminescence of Lanthanide-Doped Nanocrystals through Interstitial H⁺ doping” reports a new crystal engineering approach for enhancing the luminescence of lanthanide-doped upconversion nanoparticles. By alleviating the parity-forbidden selective rule of Ln³⁺ 4f-4f radiative transition by interstitial H⁺-doping, the authors achieved a three orders of magnitude emission enhancement in the orthorhombic phase NaMgF₃:Ln³⁺. Although the results are interesting, some problems should be first addressed to fulfil the publication requirements for Nature Communications, referring to my comments below.

Response:

Thank you for the positive comments and thoughtful suggestions, which were useful in improving the quality of our manuscript. The following is a point-by-point response to each comment.

Comment #1:

Under high temperature and vacuum conditions, acetic acid exhibits relatively high volatility. To provide more clarity on the preparation procedure, the authors should specify parameters such as the vacuum time and temperature. It is recommended that a control experiment should be conducted to determine the effect of varying vacuum time and temperature on H⁺ doping.

Response:

Thank you for this suggestion. As you mention, it is necessary to protect the reaction solvent (such as oleic acid and trioctylamine) from oxidation by air, especially given the high reaction temperature (310 °C). Furthermore, to ensure that the HAc in the reaction flask does not evaporate too quickly at the high temperature of 310 °C, the reaction device must be sealed completely, with minimal gas flow. To achieve this, we used a nitrogen atmosphere, not vacuum conditions. We apologize that this was unclear in the original manuscript. We have emphasized that the reaction was conducted in a nitrogen atmosphere in the revised manuscript and included photographs of the reaction device in the newly added Supplementary Fig. 4.

The reaction temperature and nitrogen flow rate were optimized by measuring the luminescence intensity and corresponding sample morphology of NMF-H-14.7 NCs formed under different synthesis temperatures and nitrogen flow rates. The results showed that the strongest UCL intensity and the best morphology of NMF-H-14.7 NCs were obtained at a synthesis temperature of 310 °C and a nitrogen flow rate of 0.1 L/min. See newly added Supplementary Fig. 5.

Revised text

page 3, lines 93–97: Motivated by these positive DFT results, we synthesized a series of interstitially H⁺-doped NaMgF₃:Yb/Er NCs using a modified high-temperature coprecipitation method in a nitrogen atmosphere, whereby H⁺ ions were intentionally introduced by adding acetic acid (HAc) to the reaction environment (see Methods and Supplementary Figs. 4 and 5).

New supplementary figures

Supplementary Fig. 4 | Photographs of the reaction device. To prevent the leakage of overheated acetic acid (HAc) vapor during the reaction process, it is crucial to completely seal the connection between the thermocouple (thermometer) and thermometer casing. Therefore, we sequentially wrapped the connection with parafilm, Teflon tape, and fluorinated ethylene propylene (FEP) tape. We also used a gas flowmeter to control the nitrogen (N_2) flow rate to prevent the HAc vapor from being blown out.

Supplementary Fig. 5 | **a**, UCL intensity changes and **b**, corresponding transmission electron microscopy (TEM) images of NMF-H-14.7 NCs (H^+ source: HAc; Na^+ source: NaOH) synthesized at different nitrogen (N_2) flow rates and reaction temperatures. The experimental results showed that the strongest UCL intensity and best morphology of the NMF-H-14.7 NCs were obtained at a synthesis temperature of 310 °C and N_2 flow rate of 0.1 L/min. H^+ source: HAc; Na source: NaOH. Scale bars in

b: 50 nm.

Comment #2:

Although the orthorhombic NaMgF₃ has been chosen as a model host for demonstration, the relatively low rare earth doping concentration (less than 10%) and low emission efficiency diminish the persuasiveness of the results. It is important to investigate whether this strategy can be applied to other, more efficient host matrices, such as CaF₂, cubic-NaYF₄ and hexagonal NaYF₄.

Response:

Thank you very much for your valuable suggestion. We have performed additional experiments using CaF₂ and NaYF₄. As expected, the interstitial H⁺-doping strategy greatly enhanced the luminescence intensity of CaF₂ and NaYF₄. For CaF₂, a UCL intensity enhancement factor of 807.9 was achieved for CaF₂:Yb/Er (20/2 mol%) NCs, while the corresponding upconversion quantum yield (UCQY) increased from 0.015% to 0.514%. (See newly added Supplementary Fig. 33 and Table 8).

For the cubic NaYF₄ phase, α -NaYF₄:Yb/Er (20/2 mol%) NCs with small amounts of interstitial H⁺-doping (nominal HAc addition of ≤ 3.1 mmol) achieved a UCL enhancement factor of 399, and the corresponding UCQY increased from 0.037 to 0.16%. However, unlike CaF₂, NaYF₄:Yb/Er undergoes a transition from the cubic phase to the hexagonal phase as the HAc addition was further increased. We speculate that the increase in interstitial atomic defects within the crystal lattice caused by extensive H⁺ doping may have caused the unstable cubic phase, α -NaYF₄, to transition to the more stable hexagonal phase, β -NaYF₄ (*Adv. Funct. Mater.* **2007**, *17*, 2757–2765). Nevertheless, for the completely transformed hexagonal phase, increasing the amount of HAc further continued to enhance the UCL intensity of β -NaYF₄:Yb/Er by 10.4-fold, resulting in β -NaYF₄:Yb/Er NCs with a UCQY of up to 2.601%. At present, the work on interstitial H⁺-doping of CaF₂ and NaYF₄ is still in the early stages, but we believe that after further refinement of the experimental conditions, it will be possible to obtain interstitially H⁺-doped CaF₂ and NaYF₄ NCs with higher photoluminescence quantum yields (PLQYs). (See newly added Supplementary Fig. 34 and Table 8).

Revised text

page 7–8, lines 235–244: Importantly, the interstitial H⁺-doping strategy was also suitable for other host matrices with high symmetry, such as CaF₂ and cubic α -NaYF₄, with overall UCL intensity enhancement factors of 807.9 and 399 for CaF₂:Yb/Er and α -NaYF₄:Yb/Er, respectively (Supplementary Figs. 33 and 34). The UCQYs increased from 0.015 and 0.037 to 0.514 and 0.16, respectively (Supplementary Table 8). However, the high-temperature metastable cubic phase of α -NaYF₄:Yb/Er changes to the high-temperature stable hexagonal phase when using a large nominal amount of HAc (Supplementary Fig. 34), which is likely due to the large number of interstitial atomic defects⁹. Interestingly, after complete conversion to hexagonal β -NaYF₄:Yb/Er, the continued addition of HAc to 19.2 mmol enabled a 10-fold increase in UCL intensity to ultimately obtain bright β -NaYF₄:Yb/Er NCs with a UCQY of 2.6% (Supplementary Fig. 34 and Table 8).

New added reference

46. Liang, B. X., Wang, X., Zhuang, J. & Peng, Q., Li, Y. Synthesis of NaYF₄ nanocrystals with predictable phase and phase. *Adv. Funct. Mater.* **17**, 2757–2765 (2007).

New supplementary figures and table

Supplementary Fig. 33 | **a**, Powder XRD patterns and **b**, typical UCL spectra of CaF-H-0 reference and interstitially H⁺-doped CaF₂:Yb/Er (Yb/Er = 20/2 mol%) (CaF-H-X) NCs as a function of the nominal amount of HAc used in the synthesis procedure ($X = 0-7.3$ mmol HAc). The inset in **b** showed the corresponding UCL enhancement factors. All the diffraction peaks matched well with the standard pattern of cubic-phase CaF₂ (JCPDS No. 87-0971), indicating that highly crystalline CaF₂ NCs can be formed with different interstitial H⁺-doping concentrations. Moreover, the UCL intensity of CaF₂:Yb/Er increased with the nominal addition of HAc. The maximum UCL enhancement factor was 807.9 at 7.3 mmol HAc.

Supplementary Fig. 34 | **a**, Powder XRD patterns and **b**, typical UCL spectra of NYF-H-0 reference and interstitially H⁺-doped NaYF₄:Yb/Er (Yb/Er = 20/2 mol%) (NYF-H-X) NCs as a function of the nominal amount of HAc used in the synthesis procedure ($X = 0-19.2$ mmol HAc). The inset in **b** showed the

corresponding UCL factors. The maximum UCL enhancement factors were 399 for cubic α -NaYF₄ and 10.4 for hexagonal β -NaYF₄.

New supplementary table and note

Supplementary Table 8 | Upconversion quantum yield (UCQY) of CaF-H- X ($X = 0, 3.1$ and 7.3 mmol HAc) and NYF-H- X ($X = 0, 3.1, 7.3$ and 19.2 mmol HAc) NCs.

Parameter	CaF-H- X samples			NYF-H- X samples			
	CaF-H-0	CaF-H-3.1	CaF-H-7.3	NYF-H-0	NYF-H-3.1	NYF-H-7.3	NYF-H-19.2
UCQY (%)	0.015	0.122	0.514	0.037	0.160	0.950	2.601

New Supplementary Note: Application of interstitial H⁺-doping strategy to cubic CaF₂, cubic NaYF₄, and hexagonal NaYF₄

The interstitial H⁺-doping strategy successfully enhanced the UCL intensity of cubic CaF₂ (**Supplementary Fig. 33**). A UCL intensity enhancement factor of 807.9 was achieved for CaF₂:Yb/Er (20/2 mol%) NCs, while the corresponding UCQY increased from 0.015% to 0.514% (**Supplementary Table 8**).

The interstitial H⁺-doping strategy was also applied to cubic NaYF₄ (α -NaYF₄) (**Supplementary Fig. 34**). With nominal HAc additions of less than 3.1 mmol, NaYF₄:Yb/Er retained its standard cubic phase (JCPDS No. 06-0342), with a maximum UCL enhancement factor of 399 (UCQY increased from 0.037% to 0.16%). Because the cubic phase α -NaYF₄ was not stable at high temperatures, it was easily transformed into a hexagonal phase, β -NaYF₄, under external interference at high reaction temperatures. At large interstitial H⁺-doping contents (above 3.1 mmol HAc), the cubic-to-hexagonal transition occurs (**Supplementary Fig. 34**). Nevertheless, when α -NaYF₄ was completely transformed to β -NaYF₄ (JCPDS No. 16-0334), the interstitial H⁺ ions still exerted a crystal-field perturbation effect on β -NaYF₄:Yb/Er, enhancing the UCL intensity by a factor of 10.4 (UCQY increased from 0.950% to 2.601%)

Comment #3:

Due to the small ionic radius of H⁺, it is uncertain whether it will remain stable in the interstitial site, particularly during ultrasonic dispersion. Moreover, if it is stable, the long-term stability (more than one month) should be demonstrated.

Response:

Thank you for your comment. As per your suggestion, we evaluated the stability by dispersing NMF-H-14.7 NCs in cyclohexane and deionized water (DSPE-PEG2000-modified; concentration: 3 mg/mL) and recording the change in fluorescence intensity every 2nd day for 36 days. During the 36-day continuous

observation period, no obvious changes in the fluorescence intensity was found. We speculate that the strong electronegativity of F^- led to the stable fixation of H^+ in the $NaMgF_3:Yb/Er$ crystal.

Revised text

page 8, lines 249–251: Moreover, these interstitially H^+ -doped NCs have good crystal stability, demonstrating their suitability for use in complex biological environments (Supplementary Fig. 35).

New supplementary figure

Supplementary Fig. 35 | Long-term photostability of NMF-H-14.7 NCs in **a**, cyclohexane and **b**, deionized water (DSPE-PEG2000-modified). Plots are the mean of three sets of parallel experiments and the error bars represent one standard deviation. The concentration of the solution was 3 mg/mL, and the UCL signal was collected using a continuous 980 nm NIR diode laser with a power density of ~ 100 W/cm² every second day for 36 days. The overall UCL intensity remained almost constant during continuous monitoring over a period of 36 days, strongly suggesting that the as-synthesized NMF-H-14.7 NCs have good stability in cyclohexane as well as in water. The extremely strong electronegativity of F is expected to effectively stabilize the interstitial H^+ ions within the $NaMgF_3:Yb/Er$ crystals.

Comment #4:

As the amount of HAc added greatly exceeds that of NaOH, the solution environment has undergone significant changes. It is unclear whether this affects the surface coverage of the ligand on the nanocrystals.

Response:

Thank you for this interesting comment. To explore this, we studied the NMF-H-0, NMF-H-3.1, and NMF-H-14.7 (with HAc) NCs by Fourier transform infrared (FTIR) spectroscopy, which would show any change in the ligands on the NC surface after the addition of HAc. The characterization results showed that samples NMF-H-14.7 and NMF-H-3.1 had similar surface environments to the NMF-H-0 NCs, with characteristic peaks of oleic acid ligands still present (see newly added **Supplementary Fig. 18a**).

To further exclude the influence of surface ligands, we also used acid washing (surface oleic acid ligand protonation) to remove all ligands from the surface of the NCs. The UCL intensity of the bare core state NMF-H-14.7 and NMF-H-3.1 NCs were still higher than that of NMF-H-0 (see newly added **Supplementary Fig. 18b**), indicating that changes in the surface ligands of the NCs do not have a significant impact on the implementation of the interstitial H⁺-doping strategy. This further confirms that the interstitial H⁺ doping strategy mainly affects the internal state of the crystal rather than the external environment.

To eliminate the influence of NaOH, it is possible to use NaAc instead of NaOH as the Na⁺ source; again this did not affect the interstitial H⁺-doping strategy of the NaMgF₃:Yb/Er crystals (see newly added **Supplementary Fig. 17a**).

Revised text

page 5–6, lines 168–173: Notably, introducing H⁺ ions to the NMF-H-X NCs synthesized with NaAc resulted in a gradual increase in UCL intensity (Supplementary Fig. 17a), although the synthesized samples had a worse crystal morphology than those synthesized with NaOH (Supplementary Fig. 17b). The luminescence of the NMF-H-X NCs synthesized with NaOH was also not affected by acid washing to remove surface ligands (Supplementary Fig. 18).

New supplementary figures

Supplementary Fig. 18 | a, Fourier-transform infrared (FTIR) spectra of NMF-H-0, NMF-H-3.1, and NMF-H-14.7 NCs (H⁺ source: HAc; Na⁺ source: NaOH). All three spectra contain characteristic

absorption peaks of oleic acid at 2927.8 and 2856.4 cm^{-1} (stretching vibrations of $-\text{CH}_2$) and 1560.3 and 1438.8 cm^{-1} (stretching vibrations of $-\text{COO}-$), indicating that the NMF-H-0, NMF-H-3.1, and NMF-H-14.7 NCs all have similar crystal surface environments. **b**, FTIR spectra of NCs after acid washing. The characteristic absorption peaks of oleic acid disappear, but characteristic peaks of water molecules (at 3444.7 and 1637.5 cm^{-1}) are present, indicating that the oleic acid ligands are completely removed and replaced by a layer of adsorbed water molecules. **c**, UCL spectra of NMF-H- X ($X = 0-14.7$ mmol HAc) NCs after acid washing and **d**, corresponding UCL enhancement factors. After the complete removal of surface ligands, the emission intensity of the NMF-H- X NCs still increased as the interstitial H^+ content increased, indicating that changes in the surface ligands do not have a significant impact on the implementation of the interstitial H^+ -doping strategy.

Supplementary Fig. 17 | a, Typical UCL spectra of $\text{NaMgF}_3:\text{Yb}/\text{Er}$ (with NaAc) NCs as a function of the nominal amount of HAc ($X = 0-14.7$ mmol) added during synthesis. The inset shows the corresponding UCL enhancement factors for NMF-H- X (with NaAc and HAc). The increasing UCL intensity with HAc content indicates that changing the Na^+ source does not affect the implementation of the interstitial H^+ doping strategy. **b**, TEM images of NMF-H-0 NCs synthesized with NaAc (left) and NaOH (right). Although the interstitial H^+ doping strategy can also be achieved by using NaAc as the Na^+ source, good crystal morphology cannot be obtained. Scale bars in **b**: 50 nm.

Comment #5:

After the addition of HAc, the particle size of nanoparticles increased from 12 nm to 18 nm, leading to a substantial reduction in specific surface area. This raises the question of whether the primary factor

responsible for luminescence enhancement is H^+ doping, or if other factors, such as reduced surface area, play a more prominent role. To better understand these effects, the author should attempt to synthesize nanoparticles with similar sizes for comparison.

Response:

Thank you for your valuable suggestion. NMF-H-0 NCs with particle sizes of 12, 16, and 18 nm were synthesized by adjusting the reaction temperature and reaction time, and the UCL intensities were compared with those of NMF-H-3.1, NMF-H-7.3, and NMF-H-14.7 NCs with similar particle sizes. Fluorescence spectroscopy results show that the UCL intensities of the NMF-H-3.1, NMF-H-7.3, and NMF-H-14.7 NCs were 85, 245.6, and 159.8 times brighter, respectively, than NMF-H-0 nanocrystals with an equivalent particle size. This indicates that the interstitial H^+ -doped samples have stronger UCL emission at the same particle size. (See newly added **Supplementary Fig. 11**).

Revised text

page 4–5, lines 136–140: For NMF-H-3.1 and NMF-H-0 NCs, which had a similar particle size (~12 nm), interstitial H^+ -doping increased the fluorescence intensity by a factor of 85. In addition, the fluorescence intensities of NMF-H-7.3 and NMF-H-14.7 NCs were 245.6 and 159.8 times greater than those of NMF-H-0 samples with equivalent particle sizes (~16 and 18 nm, respectively; Supplementary Fig. 11).

The newly added **Supplementary Fig. 11** is shown below

Supplementary Fig.11 | a, UCL enhancement factors of interstitially H^+ -doped $NaMgF_3:Yb/Er$ NCs relative to NMF-H-0 (without H^+ -doping) samples with a similar NC size (Na^+ source: NaOH). **b**, TEM images of non-interstitial H^+ -doped $NaMgF_3:Yb/Er$ NCs with different sizes (as listed at the bottom of each panel in **b**) synthesized at different reaction temperatures and times (as listed at the top of each panel in **b**). TEM images of interstitial H^+ -doped $NaMgF_3:Yb/Er$ NCs with different sizes (as listed at the bottom of each panel in **c**) as a function of the nominal amount of HAc used in the synthetic procedure ($X = 3.1, 7.3, 14.7$ mmol). Scale bars in **b** and **c**: 50 nm.

Comment #6:

Figure 3d reveals that the emission intensity of NMF-H-14.7 decreases after shelling, which is interesting. However, it is unclear whether this effect would have negative implications for nanoparticle applications, since coating the shell typically leads to a significant reduction in surface defect density. The author should attempt to fabricate the shell with H^+ for comparison, with similar concentrations of H^+ in both the core and the shell to suppress diffusion.

Response: Thank you for this valuable suggestion. Following your suggestion, we coated NMF-H-7.3 NCs with a H^+ -doped $NaMgF_3$ shell (same H^+ content in the core and shell) with the aim of reducing H^+ diffusion from the NC core to the inert shell. The results showed that a thin shell layer did enhance the UCL intensity of the NMF-H-7.3 NCs (see newly added **Supplementary Fig. 15**). Such doping H^+ of the shell layer is a highly complex task, as it is necessary to control the concentration of H^+ within the shell layer to prevent the diffusion of H^+ from the core, as well as to prevent fluorescence quenching caused by lattice defects due to a high H^+ concentration. At present, we only achieved a small increase in luminescence intensity through this core–shell strategy. Nevertheless, we believe that by continuing to improve the core–shell strategy, it will be possible to achieve a substantial increase in the fluorescence intensity of H^+ -doped $NaMgF_3:Yb/Er$ NCs.

Revised text

page 5, lines 155–161: To explore this hypothesis, we prepared core–shell NMF-H-7.3@ $NaMgF_3:H$ NCs in which a similar concentration of H^+ was doped into the $NaMgF_3$ shell as that in the NMF-H-7.3 core, with the aim of suppressing the diffusion of H^+ -ions from the H^+ -doped core to the pure $NaMgF_3$ shell. The UCL intensity of the NMF-H-7.3 NCs was enhanced by a thin H^+ -doped $NaMgF_3$ shell (Supplementary Fig. 15), corroborating the hypothesis that the reduction in UCL intensity with increasing shell thickness for the NMF-H-14.7@ $NaMgF_3$ NCs was due to H^+ -ion diffusion. This further illustrates the important role of H^+ in $NaMgF_3:Yb/Er$ crystals.

New supplementary figure

Supplementary Fig. 15 | a, Typical UCL spectra and **b**, TEM images of the NMF-H-7.3 and NMF-H-7.3@ $NaMgF_3:H$ core–shell NCs as a function of the thickness of the H^+ -doped $NaMgF_3$ shell (scale bar: 50 nm). The inset in **a** showed the normalized changes in UCL intensity of NMF-H-7.3@ $NaMgF_3:H$ NCs with different shell thicknesses upon continuous 980 nm NIR diode laser excitation at a power density

of $\sim 50 \text{ W/cm}^2$. The values in the bottom left corner of each panel of **b** are the mean crystal size \pm standard deviation obtained by measuring the size of 100 NCs from typical TEM images. Scale bars in **b**: 50 nm.

Thank you again for your careful review and constructive comments and advice for improving our manuscript. We would be glad to respond to any further questions and comments that you may have, or make further changes, if required.

Reviewer #4

General Comment:

In this paper, H⁺-doping in NaMgF₃:Yb/Er nanocrystals (NCs) were prepared with different contents of H⁺ and a three-orders-of-magnitude upconversion luminescence was detected under 980 nm excitation. Authors believes that the interstitial H⁺ ions perturb the local charge density distribution and thus the crystal field around the highly symmetric Ln³⁺-substituted [MgF₆]⁴⁻ octahedron cluster, which alleviates the notorious parity-forbidden selective rule to enhance the 4f–4f radiative transition rate of the Ln³⁺ emitters. It is very interest to find a new way to enhance the upconversion luminescence by doping H⁺ ions. I recommend it to be accepted after major revisions.

Response:

Thank you for the positive comments and thoughtful suggestions, which were useful in improving the quality of the manuscript. The following is our point-by-point responses to each comment.

Comment #1:

It should be clear what is the results when H⁺ content is more than 14.7mmol? and why the particle size decreases firstly and then increases along with the increasing content of HAc?

Response:

We first apologize for our unclear writing or data presentation in the original manuscript. The luminescence intensity of the NaMgF₃:Yb/Er NCs reached a maximum when the nominal amount of HAc was 14.7 mmol, and further increased in the amount of HAc led to a decrease in the luminescence intensity and change of crystal morphology (see newly added Supplementary Fig. 9 and Supplementary Note below). Because interstitial H⁺ acts as an interstitial atomic defect, excessive H⁺ doping leads to lattice distortion (e.g., changes to the lattice fringes in Supplementary Fig. 9c). Severe lattice distortion can cause fluorescence quenching and a reduction in luminescence intensity. Therefore, only a small amount of HAc (≤14.7 mmol) was added to the reaction solvent to illustrate the crystal field perturbation effect.

Regarding the comment “*why the particle size decreases firstly and then increases along with the increasing content of HAc?*”, we believe this is related to variations in the NaMgF₃:Ln³⁺ formation energy as the amount of interstitial H⁺ changes. DFT calculations showed that the formation energy increases slightly for small amounts of interstitial H⁺ doping, but further increased in the H⁺ content result in a significant decrease in the formation energy. Higher formation energies mean that the substances are more difficult to form; therefore, after the initial increase in formation energy for small amounts of interstitial H⁺ doping, NMF-H-1.6 NCs (which have a higher formation energy) have a smaller particle size than NCs with higher amounts of interstitial H⁺ (which have a lower formation energy), assuming the reaction time stays constant (as was the case here). This discussion has been added to the manuscript as a new Supplementary Note (“Crystal size of NaMgF₃ NCs with different HAc precursor additions”, page 10, lines 135–143).

New supplementary figure and note

Supplementary Fig. 9 | **a**, Typical UCL spectra, **b**, TEM images, and **c**, corresponding high-resolution TEM (HRTEM) images of as-synthesized NaMgF₃:Yb/Er NCs with excess HAc added during the synthetic procedure (14.7–25 mmol) (Na⁺ source: NaOH). The inset in **a** shows the UCL intensity as a function of HAc content. The TEM images in **b** show that excessive interstitial H⁺ doping alters the NaMgF₃:Yb/Er NC morphology. The Fourier-filtered HRTEM images in the insets in **c** reveal that excessive interstitial H⁺ doping leads to serious lattice distortion, which may be the main reason for the decrease of crystal fluorescence intensity.

New Supplementary Note: Lattice distortion at high interstitial H⁺-doping contents

The luminescence intensity of NaMgF₃:Yb/Er NCs reached a maximum when the nominal amount of HAc was 14.7 mmol, and further increased in the HAc addition led to a decrease in the luminescence intensity and change of crystal morphology (**Supplementary Fig. 9**). Because interstitial H⁺ ions were essentially interstitial atomic defects, large amounts of H⁺ doping led to lattice distortion (e.g., changes in the lattice fringes, as shown in **Supplementary Fig. 9c**). Severe lattice distortion can cause fluorescence quenching and a reduction in luminescence intensity. Therefore, only a small amount of HAc (≤ 14.7 mmol) was added to the reaction solvent to illustrate the crystal field perturbation effect.

New Supplementary Note: Crystal size of NaMgF₃ NCs with different HAc precursor additions

The crystal size of the interstitially H⁺-doped NaMgF₃ NCs first decreased and then increased with increasing interstitial H⁺ content (**Supplementary Fig. 6**). This may be related to the variation of the formation energy (ΔE_{form}) of NaMgF₃:Ln³⁺ with the amount of interstitial H⁺. DFT calculations showed

that ΔE_{form} increases slightly with small amounts of interstitial H^+ doping, but further increases in the amount of interstitial H^+ doping result in a significant decrease in ΔE_{form} (Fig. 1c). Higher ΔE_{form} values mean that the corresponding substances are more difficult to form; therefore, in the same reaction time, NCs with a higher ΔE_{form} (e.g., NMF-H-1.6) form with a smaller particle size than those with a lower ΔE_{form} (e.g., NCs with increased amounts of interstitial H^+).

Comment #2:

In fig3, it should be clear the corresponding energy level (or the corresponding wavelength) for the lifetime measurement, in addition the UCL intensity to which energy level's emission. Have you considered the emission from $^4\text{S}_{3/2}$? Because the ratio of red to green changed with the different samples, you have to describe these figures clearly.

Comment #3:

Please give a clear explanation for the reason that the ratio of red to green UCL changed along with the doping content of H^+ ions

Response:

Thank you for your careful review of our manuscript. We will respond to these comments together. In the original manuscript, Fig. 3b monitors the variation in UCL intensity between 450 and 700 nm, and Figs. 3c and d monitor the changes of UCL intensity and fluorescence lifetime of the $^4\text{F}_{9/2} \rightarrow ^4\text{I}_{15/2}$ transition of Er^{3+} , which are described in the corresponding Supplementary Figs. 13 and 14. We have relabeled Fig. 3 in the revised manuscript.

Regarding the question “Have you considered the emission from $^4\text{S}_{3/2}$?” In the $\text{NaMgF}_3:\text{Yb}/\text{Er}$ system, the UCL from $^4\text{S}_{3/2}$ was very weak, so it was not considered in the original manuscript. In fact, as the interstitial H^+ -doping concentration increases, the UCL from the $^4\text{S}_{3/2}$ state is enhanced, similarly to that from the $^4\text{F}_9$ state. We have added the data for the $^4\text{S}_{3/2}$ emissions to the manuscript. See newly added Supplementary Fig. 10.

Regarding the comment “Because the ratio of red to green changed with the different samples, you have to describe these figures clearly.” We apologize for our unclear description of the relevant diagrams in the original manuscript. The change in the red-to-green UCL ratio was related to the fine-tuning of the local symmetry of the crystal structure by interstitial H^+ -doping. We calculated the standard deviation (SD) of the bond lengths and angles of the $[\text{LnF}_6]^{3-}$ octahedral clusters with different numbers of interstitial H^+ ions to understand the distortion of the $[\text{LnF}_6]^{3-}$ octahedra (see newly added Supplementary Fig. 25). The results showed that the dispersion of the bond lengths increases only slightly as the number of interstitial H^+ ions increased from zero to two (SD increases by 0.006 Å), indicating that the introduction of H^+ has a limited effect on the dispersion of bond lengths. The eight Ln–F bonds can therefore be considered to have equal lengths. Conversely, the SD of the bond angles (F–Ln–F) was smaller with one or two interstitial H^+ ions (mean \pm SD of $90^\circ \pm 1.41^\circ$ and $90^\circ \pm 2.02^\circ$, respectively, compared to a SD of 2.06° for undoped H^+). This indicates that when the number of interstitial H^+ ions is one or two, the $[\text{LnF}_6]^{3-}$ clusters are more inclined to form perfect octahedra with high local structural symmetry. Under such high-symmetry conditions, the red-to-green UCL ratio of Er^{3+}

will increase slightly (*Nanoscale*, **2018**, *10*, 9353–9359). Therefore, in the experiment, the red-to-green ratio of samples NMF-H-2.6 and NMF-H-3.1 will be slightly higher than that with higher interstitial H⁺ contents (see revised **Supplementary Fig. 8**). This discussion has been added to the manuscript as a **new supplementary note** (“**Effect of interstitial H⁺ on sublattice structure of [LnF₆]³⁻**,” pages 37–38, lines 407–420).

Revised text

page 4, lines 132–134: Indeed, the overall UCL intensity of the NMF-H-14.7 NCs was up to 675 times higher than that of NMF-H-0 (Fig. 3b and Supplementary Fig. 9), with green and red UCL intensity enhancement factors of 258 and 706.3, respectively (Supplementary Fig. 10).

Page 7, lines 215–217: Additionally, first-principles calculations demonstrated that interstitial H⁺ only perturbs the bond angle F–Ln–F of the [YbF₆]³⁻ sublattice structure and the bond length of Yb–F remains unchanged (Supplementary Table 7 and Supplementary Figs. 25)..

New added supplementary reference

15. Fu, H. et al. A general strategy for tailoring upconversion luminescence in lanthanide-doped inorganic nanocrystals through local structure engineering. *Nanoscale* **10**, 9353–9359 (2018).

Revised Figure 3

New supplementary figures and note

Supplementary Fig. 10 | a, Green and **b**, Red UCL spectra (450–600 nm) of NaMgF₃:Yb/Er NCs synthesized with HAc (nominal amounts of 0–14.7 mmol) (Na⁺ source: NaOH) under 980 nm diode laser excitation with a power density of 50 W/cm². The insets in **a** and **b** showed the corresponding enhancement factors of the green and red UCL intensities for NMF-H-X (with HAc). The interstitial H⁺ doping strategy can effectively enhanced the green and red UCL intensities simultaneously, with enhancement factors of up to 258 and 706.3 for green ($^2H_{11/2}, ^4S_{3/2} \rightarrow ^4I_{15/2}$) and red ($^4F_{9/2} \rightarrow ^4I_{15/2}$) emissions, respectively.

Supplementary Fig. 25 | Distribution of **a**, Yb–F bond lengths and **b**, F–Yb–F bond angles within $[\text{YbF}_6]^{3-}$ octahedral clusters with different numbers of interstitial H^+ ions (0–3 H_i), as determined by first-principles DFT calculations. Because lanthanide ions have similar atomic structures, Yb was used as a representative lanthanide ion. The mean \pm standard deviation bond lengths and angles are displayed in the upper right corner of each panel.

Supplementary Note: Effect of interstitial H^+ on sublattice structure of $[\text{LnF}_6]^{3-}$

Supplementary Fig. 25 showed that the dispersion of the bond lengths increases only slightly as the number of interstitial H^+ ions increased from zero to two (standard deviation increases by 0.006 Å), indicating that interstitial H^+ -doping has a limited effect on the dispersion of bond lengths. The eight Ln–F bonds can therefore be considered of equal length. Conversely, the standard deviation of the bond angles (F–Ln–F) was smaller with one or two interstitial H^+ ions (mean \pm standard deviation of $90^\circ \pm 1.41^\circ$ and $90^\circ \pm 2.02^\circ$, respectively, compared to a standard deviation of 2.06° for undoped H^+). This indicates that when the number of interstitial H^+ ions is one or two, the $[\text{LnF}_6]^{3-}$ clusters are more inclined to form perfect octahedra with high local structural symmetry. Under such high-symmetry conditions, the R/G intensity ratio of Er^{3+} will increase slightly⁹. Therefore, in the experiments, the R/G ratios of samples NMF-H-2.6 and NMF-H-3.1 were slightly higher than those of the samples with higher interstitial H^+ contents (**Supplementary Fig. 8**). In addition, the limited change in the symmetry of the $[\text{LnF}_6]^{3-}$ sublattice structure at low interstitial H^+ concentrations led to similar thermal enhancement of UCL in samples NMF-H-0 and NMF-H-3.1 (**Supplementary Fig. 26**).

Supplementary Fig. 8 | Normalized emission spectra of NaMgF₃:Yb/Er NCs as a function of the nominal amount of HAc (0–14.7 mmol). (Na⁺ source: NaOH.) The inset displays the corresponding red-to-green (R/G) UCL intensity ratio. Red UCL wavelength: 615–725 nm, green UCL wavelength: 505–575 nm. As the nominal amount of HAc increases, the NaMgF₃:Yb/Er NCs maintain a high R/G UCL intensity ratio. NMF-H-2.6 and NMF-H-2.6 have higher R/G ratios than the other samples, because small concentrations of interstitial H⁺ ions may slightly improve the symmetry of the [LnF₆]³⁻ sublattice structure. Once the interstitial H⁺ concentration increases past a certain point, the symmetry of the [LnF₆]³⁻ octahedra will be weakened again (**Supplementary Fig. 25**). Overall, the interstitial H⁺ ions only have a limited effect on the [LnF₆]³⁻ sublattice structure (crystal-field perturbation); therefore, there is little change in the green UCL intensity in the normalized UCL spectra.

Comment #4:

In fig4c, it is better to change the Ln to Er (emitter). I believe that H doping also affect the crystal field around Yb, but the energy level of Yb³⁺ is very simple and the absorption of Yb³⁺ increased weaker than the upconversion luminescence of Er³⁺. Therefore, the effect of H⁺ doping on the transition rate of Er³⁺ and the distance of between Yb and Er should be discussed mainly.

Response:

Thank you for pointing this out. We have changed Ln to Er, thus giving a more accurate representation of **Fig. 4c** in the revised manuscript.

Regarding the comment “*Therefore, the effect of H⁺ doping on the transition rate of Er³⁺ and the distance of between Yb and Er should be discussed mainly,*” in order to understand the change in the Er³⁺ transition rate, we monitored the photoluminescence decays of the ⁴S_{3/2}→⁴I_{15/2} and ⁴F_{9/2}→⁴I_{15/2} transitions of Er³⁺. As the interstitial H⁺ concentration increased, the time-resolved decay curves of the ⁴S_{3/2}→⁴I_{15/2} and ⁴F_{9/2}→⁴I_{15/2} transitions both shortened at first and then prolonged (see **newly added Supplementary Fig. 22**). Comparing NMF-H-0 and NMF-H-3.1, which had a similar particle size, the NMF-H-3.1 NCs had a stronger UCL intensity but shorter UCL lifetime than the NMF-H-0 NCs. This trend was more pronounced at low temperatures (10 K; Fig. 4e). Considering that the NCs have the same size and morphology, the influence of the external environment on the nonradiative transition probability can be assumed to be approximately equal. According to the relationship between the fluorescence lifetime and the transition probability ($\tau = (A_{\text{ed}} + W_{\text{NR}})^{-1}$), the shortening of the fluorescence lifetime (τ) represents an increase in the radiative transition probability (A_{ed}), as the nonradiative transition probability (W_{NR}) is equal. Therefore, the introduction of interstitial H⁺ can simultaneously enhance the radiative transition probabilities of the ⁴S_{3/2}→⁴I_{15/2} and ⁴F_{9/2}→⁴I_{15/2} transitions of the Er³⁺ emitter. This discussion has been added to the manuscript as a new **Supplementary Note (“Effect of interstitial H⁺ doping on the Er³⁺ transition rate,”** page 28, lines 314–325).

Regarding the effect of interstitial H⁺ on the distance between Yb and Er, we conducted first-principles DFT calculations that showed that, non-interstitial H⁺-doped NaMgF₃:Yb/Er and interstitial H⁺-doped NaMgF₃:Yb/Er, the distances between adjacent Yb and Er atoms were 4.044 and 4.061 Å, respectively.

Consequently, there is little change in the atomic distance between Yb and Er. This may mean that the presence of interstitial H^+ has a limited effect on Yb-Er spacing. We therefore believe that the UCL enhancement from Er^{3+} mainly occurs because interstitial H^+ -doping exerts a crystal-field perturbation effect by inducing the anisotropic polarization of the ligand (F^-), which leads to the introduction of asymmetric elements into Er^{3+} , resulting in enhanced fluorescence intensity (see newly added **Supplementary Fig. 21**).

Revised Figure 4

New supplementary figure and note

Supplementary Fig. 22 | Time-resolved decay curves of **a**, ⁴S_{3/2} → ⁴I_{15/2} and **b**, ⁴F_{9/2} → ⁴I_{15/2} transitions of the Er³⁺ emitter in NaMgF₃:Yb/Er NCs with different nominal amounts of HAc (X = 0–14.7 mmol HAc) (Na⁺ source: NaOH). All the UCL decay curves were measured using a 980 nm pulsed laser at room

temperature. The insets show the average UCL lifetimes (τ) from fitting the decay curves with a triple exponential function.

New Supplementary Note: Effect of interstitial H⁺ doping on the Er³⁺ transition rate

The time-resolved decay curves of the $^4S_{3/2} \rightarrow ^4I_{15/2}$ and $^4F_{9/2} \rightarrow ^4I_{15/2}$ transitions of the Er³⁺ emitter (**Supplementary Fig. 22**) show that, as the interstitial H⁺ concentration increases, the UCL lifetime first shortens and then prolongs. Comparing NMF-H-0 and NMF-H-3.1, which have a similar particle size, the NMF-H-3.1 NCs have a stronger UCL intensity but shorter UCL lifetime than the NMF-H-0 NCs. This trend was more pronounced at low temperatures (10 K; Fig. 4e). When external conditions such as the size and morphology of the NCs are the same, the influence of the external environment on the nonradiative transition probability W_{NR} can be assumed to be approximately equal. Consequently, according to the relationship $\tau = (A_{ed} + W_{NR})^{-1}$, the shortening of the fluorescence lifetime τ represents an increase in the radiative transition probability A_{ed} . Therefore, interstitial H⁺-doping can simultaneously enhance the radiative transition probabilities of the $^4S_{3/2} \rightarrow ^4I_{15/2}$ and $^4F_{9/2} \rightarrow ^4I_{15/2}$ transitions of the Er³⁺ emitter.

Supplementary Fig. 21 | DFT-calculated electronic band structures and their corresponding projected DOS for **a**, Non-interstitial H⁺-doped NaMgF₃:Yb and **b**, Interstitial H⁺-doped NaMgF₃:Yb NCs. DFT-calculated transition matrix elements for **c**, Non-interstitial H⁺-doped NaMgF₃:Yb and **d**, Interstitial H⁺-doped NaMgF₃:Yb NCs.

Supplementary Note: Effect of interstitial H⁺ on band structure and dipole transition matrix

elements of NaMgF₃:Ln

DFT calculations showed that the introduction of interstitial H⁺ ions had a limited effect on the band gap, shrinking it by only 0.173 eV. However, the corresponding density of states (DOS) showed that interstitial H⁺ contributes to the valence band, which indicates that although the effect of interstitial H⁺ on the crystal structure was limited, it still caused a change in the charge distribution within the crystal (**Supplementary Fig. 21a and b**). Additionally, the effect of interstitial H⁺ ions on the transition dipole moment can be observed through DFT theoretical calculations based on first principles. Prior to the introduction of H⁺, the dipole transition matrix elements of NaMgF₃:Ln were mainly concentrated at the high-symmetry G point (**Supplementary Fig. 21c**). Therefore, the radiative transition at the 4f^N level is parity forbidden. The introduction of interstitial H⁺ produced an additional field and disturbed the original field of the crystal, resulting in the transfer of the transition dipole moment elements from the high-symmetry G point to the low-symmetry X|Y point (**Supplementary Fig. 21d**). The existence of dipole transition matrix elements with low symmetry can promote the mixing of opposite-parity configurations of lanthanide ions, thereby improving the efficiency of the Ln³⁺-emitter intra-4f optical transitions.

Thank you again for your careful review and constructive comments and advice for improving our manuscript. We would be glad to respond to any further questions and comments that you may have, or make further changes, if required.

REVIEWERS' COMMENTS

Reviewer #1 (Remarks to the Author):

The authors have well addressed all raised issues; therefore, the manuscript can be accepted for publication.

Reviewer #2 (Remarks to the Author):

The revised manuscript is improved in quality and is recommended for publication.

Reviewer #3 (Remarks to the Author):

The authors have carefully addressed all my concerns in this revised manuscript. I feel this work represents a timely and significant contribution to the field of upconversion nanomaterials. I recommend publication as is.

Reviewer #4 (Remarks to the Author):

This manuscript has been revised according all comments. I recommend to accept it.

Fujian Institute of Research on the Structure of Matter
Chinese Academy of Sciences

Response to Editor and Reviewers' Comments

Reviewer #1 (Remarks to the Author): The authors have well addressed all raised issues; therefore, the manuscript can be accepted for publication.

Reviewer #2 (Remarks to the Author): The revised manuscript is improved in quality and is recommended for publication.

Reviewer #3 (Remarks to the Author): The authors have carefully addressed all my concerns in this revised manuscript. I feel this work represents a timely and significant contribution to the field of upconversion nanomaterials. I recommend publication as is.

Reviewer #4 (Remarks to the Author): This manuscript has been revised according all comments. I recommend to accept it.

Response: We are very grateful to the reviewers for their affirmation of our work, as well as their positive comments and valuable suggestions on the manuscript.